# Reduced C9orf72 function leads to defective synaptic vesicle release and neuromuscular dysfunction in zebrafish

Zoé Butti[1], Yingzhou Edward Pan[1], Jean Giacomotto[2,3] & Shunmoogum A. Patten [1,4]✉

The most common genetic cause of amyotrophic lateral sclerosis (ALS) and fronto-temporal dementia (FTD) is a hexanucleotide repeat expansion within the *C9orf72* gene. Reduced levels of *C9orf72* mRNA and protein have been found in ALS/FTD patients, but the role of this protein in disease pathogenesis is still poorly understood. Here, we report the generation and characterization of a stable C9orf72 loss-of-function (LOF) model in the zebrafish. We show that reduced *C9orf72* function leads to motor defects, muscle atrophy, motor neuron loss and mortality in early larval and adult stages. Analysis of the structure and function of the neuromuscular junctions (NMJs) of the larvae, reveal a marked reduction in the number of presynaptic and postsynaptic structures and an impaired release of quantal synaptic vesicles at the NMJ. Strikingly, we demonstrate a downregulation of SV2a upon C9orf72-LOF and a reduced rate of synaptic vesicle cycling. Furthermore, we show a reduced number and size of Rab3a-postive synaptic puncta at NMJs. Altogether, these results reveal a key function for C9orf72 in the control of presynaptic vesicle trafficking and release at the zebrafish larval NMJ. Our study demonstrates an important role for C9orf72 in ALS/FTD pathogenesis, where it regulates synaptic vesicle release and neuromuscular functions.

[1] INRS- Centre Armand-Frappier Santé Biotechnologie, Laval, QC, Canada. [2] Queensland Brain Institute, University of Queensland, St Lucia, QLD, Australia. [3] Queensland Centre for Mental Health Research, Brisbane, QLD, Australia. [4] Centre d'Excellence en Recherche sur les Maladies Orphelines - Fondation Courtois (CERMO-FC), Université du Québec à Montréal (UQAM), Montréal, QC, Canada. ✉email: kessen.patten@inrs.ca

Amyotrophic lateral sclerosis (ALS) is a progressive and ultimately lethal neuromuscular disease involving the degeneration and loss of motor neurons. Current Food and Drug Administration-approved treatments for ALS are only modestly effective and the disease still results in complete paralysis and death within the first 5 years after diagnosis. GGGGCC hexanucleotide repeat expansions within the first intron of *C9orf72* is the most common genetic cause of ALS and frontotemporal dementia (FTD)[1,2]. The pathogenic mechanism by which the repeat expansions cause disease may involve toxic gain-of-function (GOF) mechanisms, such as RNA toxicity[3] and protein toxicity, by aberrant dipeptide repeat protein (DPR) accumulation[4,5]. Alternatively, reduced C9orf72 mRNA and protein levels in a range of patient tissues and patient-derived cell lines[1,6,7] suggest that loss of function (LOF) by *C9orf72* haploinsufficiency may also contribute to C9orf72 ALS/FTD.

The two GOF pathogenic mechanisms are extensively studied[8], while the role of C9orf72-LOF in ALS pathogenesis remains poorly understood. Importantly, in general, how the GGGGCC hexanucleotide repeat expansions cause neurodegeneration in ALS and FTD is still uncertain. The C9orf72 protein has been shown to function in a complex with the WDR41 and SMCR proteins as a guanine exchanging factor (GEF) for Rab8 and Rab39[9,10]. It has also been proposed to play a role in autophagic flux[9,11,12], endosomal trafficking[13–15] and regulating AMPA receptor levels[16].

Synaptic alterations at neuromuscular junctions (NMJs) have been found in ALS patients and in animal models of ALS. For instance, Killian et al. observed that initial compound motor action potentials (CMAP) in ALS patients were of low amplitude but did not demonstrate early post-exercise facilitation (reduction in decrement occurred at 3 min post-exercise). The low baseline CMAP amplitudes with decrement may suggest a presynaptic transmission deficit[17]. In vitro microelectrode studies of ALS patient anconeus muscle demonstrated reduced presynaptic acetylcholine quantal stores, possibly explained by the diminished size of nerve terminals[17]. In mutant *SOD1*-expressing mice[18,19] an early retraction of presynaptic motor endings was observed long before the death of motoneurons[20]. Such an observation was also observed in tissue from patients with ALS[17]. In zebrafish, expression of mutant human TARDBP[G348C] mRNA or FUS[R521H] resulted in impaired transmission, reduced frequency of miniature endplate currents (mEPCs) and reduced quantal transmission at the NMJ[21,22]. C9orf72 is expressed presynaptically and postsynaptically[16]. The function of C9orf72 at synapses remains interesting and largely unexplored, yet a full understanding of its synaptic function can extend its contribution to ALS pathogenesis and uncover therapeutic targets.

Zebrafish is a powerful tool for studying neurological diseases relevant to humans, including ALS[23]. Using a stable transgenic zebrafish model with reduced C9orf72 expression, we analysed the effects of reduced C9orf72 function on the zebrafish's neuromuscular system. These zebrafish display behavioural deficits and early mortality as observed in C9orf72-ALS patients. C9orf72 silencing resulted in impaired synaptic activity and downregulation of the synaptic protein, synaptic vesicle (SV)-associated protein 2a (SV2a). Our findings suggest that LOF mechanisms underlie defects in synaptic function in C9-ALS.

## Results

### Generation of a stable C9orf72-LOF model in zebrafish. To better understand the role of *C9orf72*-LOF in ALS/FTD pathogenesis, we generated a stable transgenic zebrafish gene-silencing model. A single conserved *C9ORF72* orthologue is present in zebrafish on its chromosome 13. To achieve transgenic *c9orf72*

gene silencing in zebrafish, we used a recent miRNA-based gene-silencing approach developed for zebrafish[24]. Unlike morpholino-based knockdown approach, transgenic zebrafish lines that have been constructed to stably express miRNAs designed to target knockdown desired genes of interest have no apparent non-specific toxic effects[25]. The miRNA knockdown technique consists in the use of transgenic DNA construct allowing the expression of synthetic miRNA targeting the 3'-untranslated region (3'-UTR) of a gene of interest, here the endogenous zebrafish *c9orf72* (Fig. 1a). As presented more in detail in the 'Methods' section, we designed 4× different miRNAs targeting specifically *c9orf72* (C9orf72-miR) that we inserted downstream of a dsRED marker and under the control of a ubiquitous promoter (Ubiquitin), and the overall sequence was recombined into a mini-Tol2-R4R2 destination plasmid. To generate a transgenic line, this Tol2-DNA construct was co-injected with transposase mRNA in fertilized eggs at the one-cell stage for enhanced genomic integration of the DNA construct[26]. To ease the selection of the founders/carriers, we also included an enhanced green fluorescent protein (eGFP) cassette under the crystallin promoter (Fig. 1a). Founders with eyes displaying GFP fluorescence (Fig. 1b) were selected and raised to generate a stable and heritable C9orf72-miR LOF line (hereafter referred as C9-miR). F1 transgenic fish gave a birth to a ratio of close to 50% positive GFP embryos when outcrossed with wild-type animals, suggesting the presence of a single genomic insertion.

We first analysed *C9orf72* silencing efficiency in our C9-miR line by reverse transcriptase quantitative PCR (RT-qPCR) and western blotting. We showed a significant decrease in the level of C9orf72 *mRNA* (Fig. 1c) associated with a 50% decrease of C9orf72 protein (Fig. 1d, e and Supplementary Fig. 1). Altogether, these results indicate that our genetic approach efficiently reduces the C9orf72 protein levels in vivo, and this C9-miR line can be used to understand the role of C9orf72 haploinsufficiency in ALS.

### C9orf72-LOF model shows early motor behavioural defects and reduced viability. We did not observe any overt morphological abnormalities during embryonic development (0–5 days post fertilization (dpf)) in C9-miR fish (Fig. 1f). From 6 to 14 dpf, some C9-miR larvae exhibited gradual morphological defects such as an unusual body curve and premature death (Fig. 1f, g). C9orf72 partial depletion importantly led to a significant decrease in survival at 10 dpf compared to wild-type controls, with a survival rate of 2–5% after 15 dpf (Fig. 1g).

We next examined whether normal zebrafish motor behaviour was affected in larval C9-miR zebrafish (4–11 dpf). To assess motor activity, larval zebrafish that did not display any of the abnormal morphological defects were selected and monitored using the automated Noldus Ethovision XT behaviour monitoring system (Fig. 2a). A significant decrease in motor activity was observed in C9-miR fish as compared to controls, as of 6 dpf (Fig. 2a, b). Such an impaired motor behaviour early on in C9-miR zebrafish is consistent with findings that we and others have reported in several other zebrafish models of ALS[23,27–29]. To test the specificity of the zebrafish C9orf72-LOF motor behavioural phenotype, mRNA encoding the human *C9orf72* long transcript (C9-rescue) was injected into C9-miR fish, which significantly rescued the motor behavioural defects (Fig. 2a, b).

### C9orf72 silencing affects NMJ structural integrity and quantal release. To understand the underlying causes of motor behavioural defects at 6 dpf in C9-miR fish, we next examined NMJ integrity by performing double immunohistochemistry at larval stages using specific presynaptic (SV2) and postsynaptic markers (α-bungarotoxin). In 6 dpf C9-miR larvae, we observed a

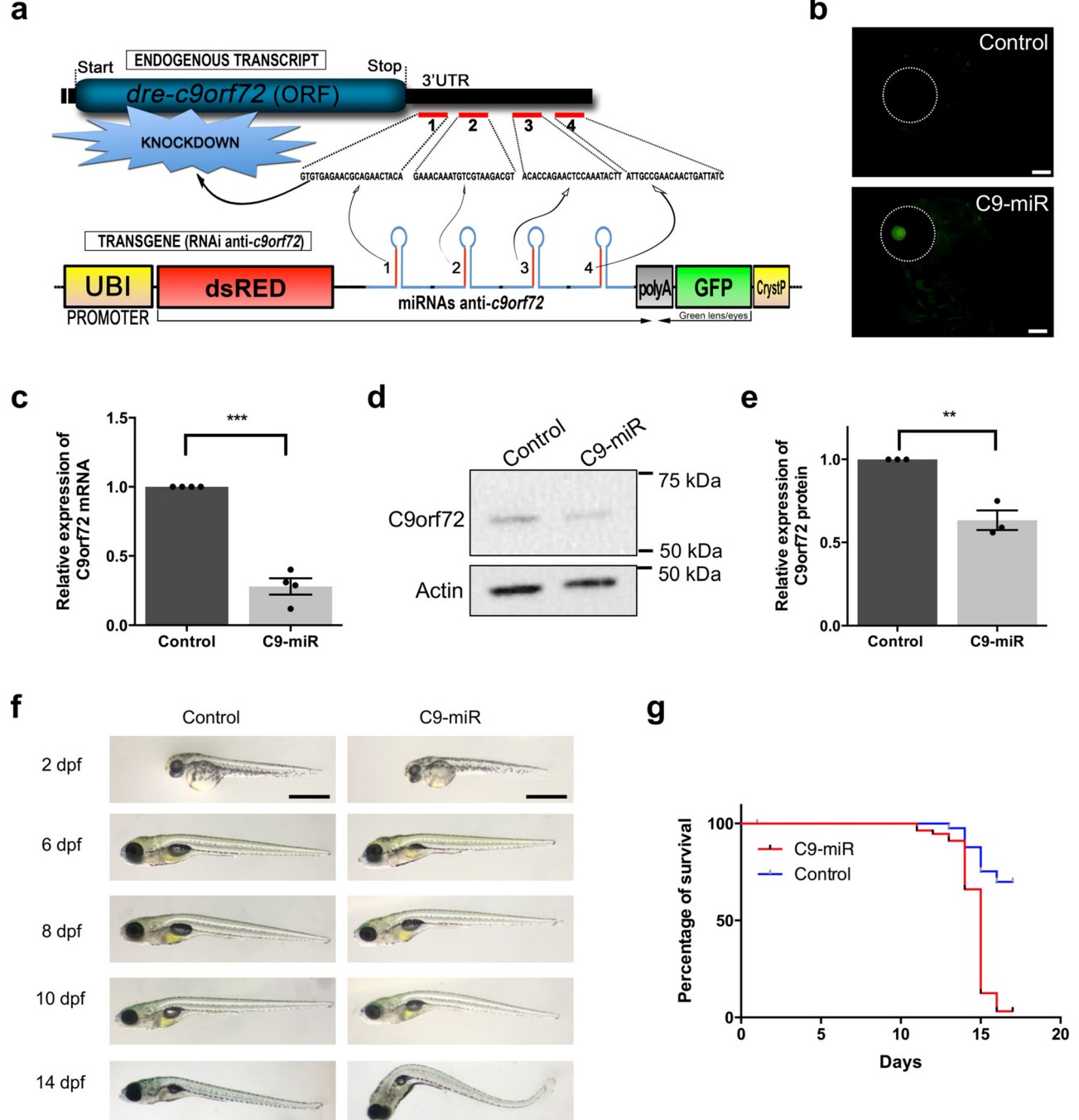

**Fig. 1 Generation of a stable zebrafish C9orf72 knockdown line. a** Schematic representation of the technique used to silence *C9orf72* in zebrafish. The transgene is designed to express four different micro-RNAs targeting C9orf72's 3'UTR and triggering knockdown by both repressing *C9orf72* translation and affecting its stability. **b** Images demonstrating proper eGFP expression in the crystallin of the transgenic fish, a marker used to identify carrier/knockdown larvae. Scale bar = 100 μm. **c** Bar graph shows the relative expression of the endogenous *C9orf72* gene. mRNA was normalized to elf1α mRNA levels ($N = 4$, ***$p < 0.0001$, Student's $t$ test). **d** Immunoblotting of the zebrafish protein C9orf72 and beta-actin as control. **e** Bar graph shows the relative expression of the C9orf72 protein compared with actin between C9orf72 mutants and control line ($N = 3$; **$p = 0.0034$; Student's $t$ test). **f** Gross morphological analyses of wild-type control and C9orf72-LOF fish (C9-miR). **g** Kaplan–Meier survival plot over 17 days after fertilization (dpf) showing low survival of C9-miR compared to controls after 10 dpf ($N = 3$, $n = 25$). Data are presented as mean ± SEM. $n$ represents the number of fish, $N$ represents the number of experimental repeats.

significant reduction in the number of colocalizing presynaptic and postsynaptic puncta (Fig. 2c, d) compared to control larvae. These NMJ anomalies were significantly rescued upon expression of human *C9orf72* long transcript (Fig. 2c, d). Analysis revealed no change in the primary motor neuron axon architecture and in the colocalization of presynaptic and postsynaptic signals in C9-

miR fish at early developmental stages (2 dpf; Supplementary Fig. 2). Altogether, these results indicate that, while the synaptic structures of the NMJ develop properly and are preserved at early larval stages in C9-miR, they do start to degenerate as of 6 dpf.

To investigate whether alterations in NMJ integrity had functional consequences on synaptic transmission in the 6 dpf

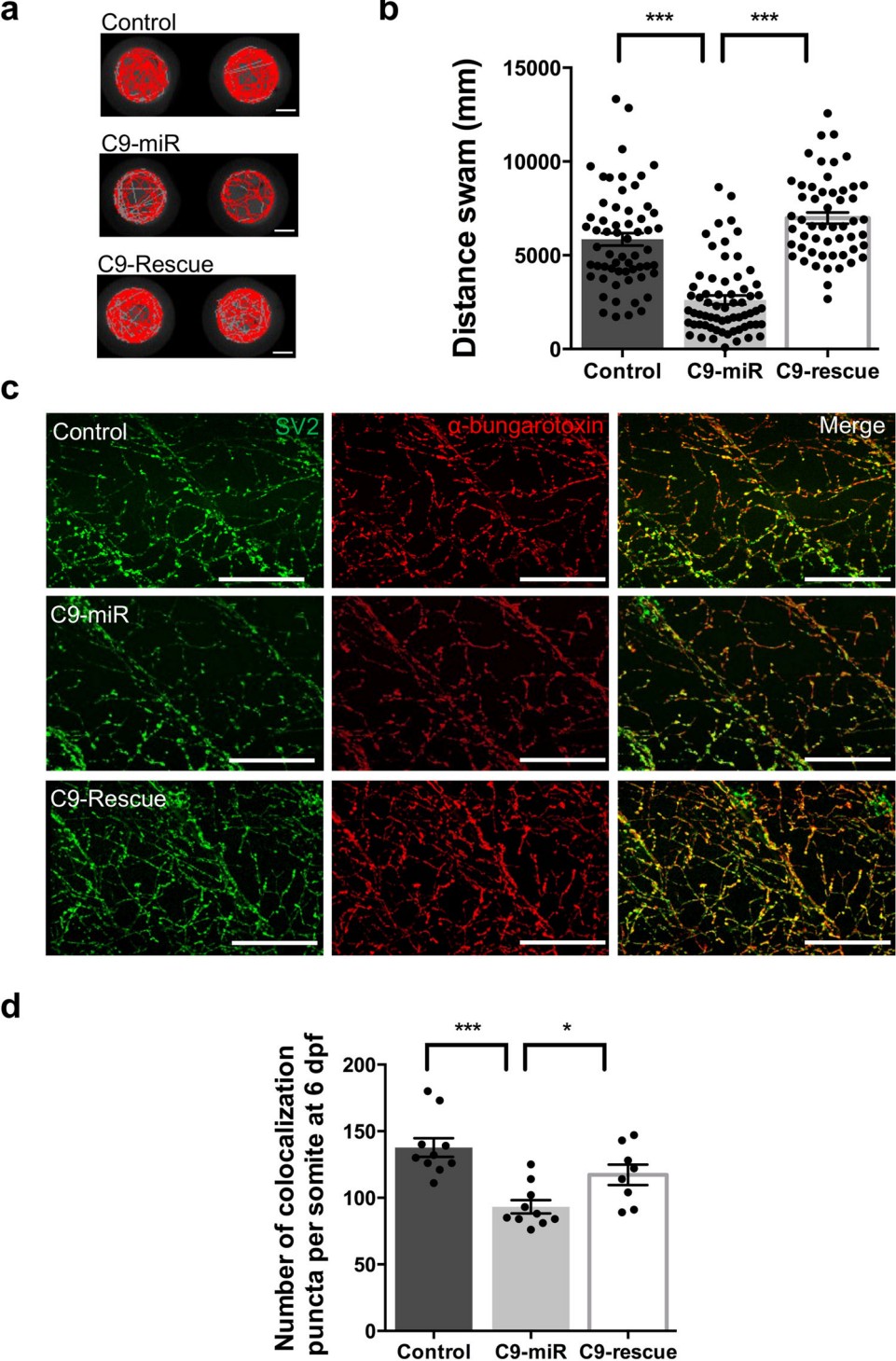

**Fig. 2 Motor behavioural deficits and reduced acetylcholine receptor clusters at neuromuscular junctions (NMJs) in C9-miR fish. a** Representative swimming tracks of control, C9-miR and C9-rescue fish at 6 dpf. Scale bar = 0.2 cm. **b** C9-miR larvae ($N = 3$, $n = 65$) displayed impaired swimming compared to controls ($N = 3$, $n = 60$; ***$p < 0.0001$; one-way ANOVA). Expression of the human *C9orf72* long transcript in C9-miR fish (C9-rescue; $N = 3$, $n = 53$) significantly rescued the motor behavioural defects (***$p < 0.0001$; one-way ANOVA). **c** Representative images of co-immunostaining of zebrafish neuromuscular junctions with presynaptic (SV2; green) and postsynaptic (α-bungarotoxin; red) markers in 6 dpf zebrafish. Scale bar = 100 μm. **d** Quantification of the colocalizing presynaptic and postsynaptic markers per somite showed a significant reduction in the number of puncta in C9-miR fish at 6 dpf that can be rescued with the expression of human *C9orf72* mRNA (C9-rescue) ($n = 8$–10; ***$p < 0.0001$; one-way ANOVA). Data are presented as mean ± SEM. $n$ represents the number of fish, $N$ represents the number of experimental repeats.

C9-miR larvae, we recorded and analysed the spontaneous mEPCs that occur naturally and spontaneously at synapses and represent the unitary event during synaptic transmission (Fig. 3a, b). We observed that the frequency of mEPCs in C9-miR was significantly reduced compared to controls (Fig. 3c), suggesting a reduction in the number of functional presynaptic endings. The mean amplitude of mEPCs was also found to be smaller in zebrafish C9-miR compared to wild-type zebrafish (Fig. 3d). We observed that the mEPCs from the muscle of C9-miR larvae and controls shared similar rise time and decay time constant kinetics (Fig. 3e, f).

**C9orf72-LOF model displays TDP-43 pathology.** Cytoplasmic aggregation of trans-activation response element (TAR) DNA-binding protein 43 (TDP-43) is a major pathological hallmark of ALS[30]. TDP-43 forms aggregates in neurons, glial cells[30] and axial skeletal muscles[31]. By taking advantage of the relatively large nucleus and cytoplasm of the skeletal muscle cells (Fig. 4a), we examined whether TDP-43 pathology exist in our model. Using a specific antibody that recognizes the highly homologous human TDP-43 orthologue in zebrafish[32], we showed that this protein is localized to the nucleus of the skeletal muscle cells at 6 dpf in wild-type zebrafish (Fig. 4a). In contrast, in C9-miR zebrafish, we observed clusters of TDP-43 that are predominantly located outside of the nucleus of muscle cells (Fig. 4b and Supplementary Fig. 2b). Quantitative analysis revealed a significant reduction of the nuclear-to-cytoplasmic (N-to-C) ratio of TDP-43 in C9-miR fish compared to controls, evidencing the cytoplasmic accumulation of TDP-43 in the muscles of C9-miR fish (Fig. 4c). This TDP-43 pathology in C9-miR zebrafish was significantly rescued upon expression of the human C9orf72 mRNA (Fig. 4b, c). Altogether, our findings provide strong evidence that C9orf72 silencing in zebrafish recapitulates a key pathological hallmark of ALS.

**C9orf72-LOF zebrafish model display adult hallmark features of ALS.** C9-miR fish that survive past 15 dpf were also studied at adult stages for hallmarks of ALS such as motor behavioural anomalies/paralysis and neuromuscular deficits. At the motor behavioural level in 12 month fish, we observed an impaired swimming ability in adult C9-miR animal compared to controls (Fig. 5a, b and Supplementary Movies 1 and 2). Prior to death, C9-miR fish spent their time in the bottom of the tank with weak movements and showing signs of paralysis (a stage that we termed 'end stage'). Adult survival was also monitored and we observed that by 16 months post-fertilization, >80–90% of the adult C9-miR zebrafish die. We next investigated neuromuscular pathology by first examining NMJ integrity in 12-month-old control and C9-miR fish (Fig. 5c). Quantification of the number of colocalized presynaptic (SV2) and postsynaptic (α-bungarotoxin) puncta revealed a significantly reduced number of synaptic puncta at NMJs in adult C9-miR fish compared to adult control fish (Fig. 5d). NMJ degeneration is followed by motoneuron loss in ALS patients and animal models[18,19]. Using choline acetyltransferase (ChAT) staining, we next examined the motor neurons in the spinal cord of C9-miR and control adult fish. Motor neurons in the zebrafish spinal cord are small or large depending on the maturation stage of the neurons and vary in physiology[33,34]. Large-sized motor neurons (≥10 μm in diameter) are fast-fatigable motor neurons that are most vulnerable to degeneration[35]. We observed an overall reduction in the total number of ChAT-positive motor neurons in C9-miR fish (Fig. 5e) and the mature motor neurons (i.e. large cell body) were reduced in size compared to control fish (Fig. 5e, f), consistent with motor neuron degeneration pathology in ALS patients.

Haematoxylin and eosin (H&E) staining of cross-section of fish body trunk revealed that the muscles in adult C9-miR exhibited severe atrophy (Fig. 5g), with a significant reduction in the thickness of the fibres (Fig. 5g, h).

Given that the nuclei of motor neurons are largein size and easily examined in adult spinal cord sections, we examined whether TDP-43 pathology occur in spinal motor neurons in C9-miR animals. Cells in the spinal sections of 14- to 16-month-old fish were stained for ChAT, TDP-43 and 4,6-diamidino-2-phenylindole (DAPI) (Fig. 6). Strikingly, we observed unusual circular clustering of TDP-43 stains that are negative for both DAPI staining in the grey matter of the spinal cord (Fig. 6a, b), indicating that TDP-43 clusters reside in the cytoplasm. Since the expression of TDP-43 is dispersed in wild-type control fish (Fig. 6a) and that neurons in the zebrafish spinal cord are in very close proximity, this complicates the quantification of the N-to-C expression level of TDP-43. Given that nuclear depletion of TDP-43 has been reported as a measure of TDP-43 pathology in several studies[36–38], we thus quantified the percentage colocalization between the cellular TDP-43 antibody signal and the nuclear DAPI stain in ChAT+ motor neurons in control and C9-miR spinal cord sections. Compared to controls, the degree of colocalization of TDP-43 signal and DAPI stain in spinal motor neurons was significantly reduced, suggesting a nuclear depletion in C9-miR motor neurons (Fig. 6a–c).

**C9orf72 regulates SV exocytosis and synapse stability at the NMJ.** To gain molecular insights into the function of C9orf72, we examined the processes and pathways affected in C9-miR fish by determining global changes at the proteomic levels by isolating total proteins at 6 dpf from C9-miR larvae and wild-type siblings. We identified a total of 2602 proteins, out of which 2093 proteins were covered by ≥2 unique peptides and were quantifiable in 4 biological replicates (false discovery rate ≤1%; Fig. 7a). Most of the proteins in wild-type and C9-miR fish were at comparable expression levels. Only 24 proteins were found to be dysregulated ($p < 0.05$; log2 fold-change of −1.5 and 1.5; Supplementary Table 2). Of these hits, 12 were upregulated and 12 were down-regulated in C9-miR fish (Fig. 7b). These differentially expressed proteins (DEPs) were classified into functional clusters according to the PANTHER classification system (Supplementary Fig. 3). The classification results revealed that many DEPs were distributed into six protein classes (Supplementary Fig. 3a). These proteins are classified into three molecular functions, namely binding (20%), structural molecule activity (20%) and catalytic activity (60%) (Supplementary Fig. 3b). They are involved in biological processes, with cellular process, metabolic process and biological regulations being the most represented ones with 38%, 23.1% and 15.4% of proteins, respectively (Supplementary Fig. 3c). Cellular component analysis revealed that the DEPs belong in majority to the organelle, membrane and synapse categories (Supplementary Fig. 3d). Consistent with the synaptic dysfunction phenotype, we identified a strong downregulation of synaptic proteins (Fig. 7b and Supplementary Table 2). Among these proteins, the top hit of dysregulated proteins is the synaptic protein SV2a. Importantly, a recent study showed that SV2a is reduced in C9orf72-ALS patient-derived induced pluripotent stem cell (iPSC) neurons[39]. This data links the findings in our C9orf72-LOF model to ALS. In order to test whether C9orf72 interacts with SV2a, we performed co-immunoprecipitation coupled with western blot. HEK293T cells were co-transfected with RFP-C9orf72 with Myc-SV2a or Myc-SMCR8, a known interaction partner of C9orf72 as positive control[13]. SMCR8 and SV2a were also co-transfected with red fluorescent protein (RFP) as negative control. We found that Myc-

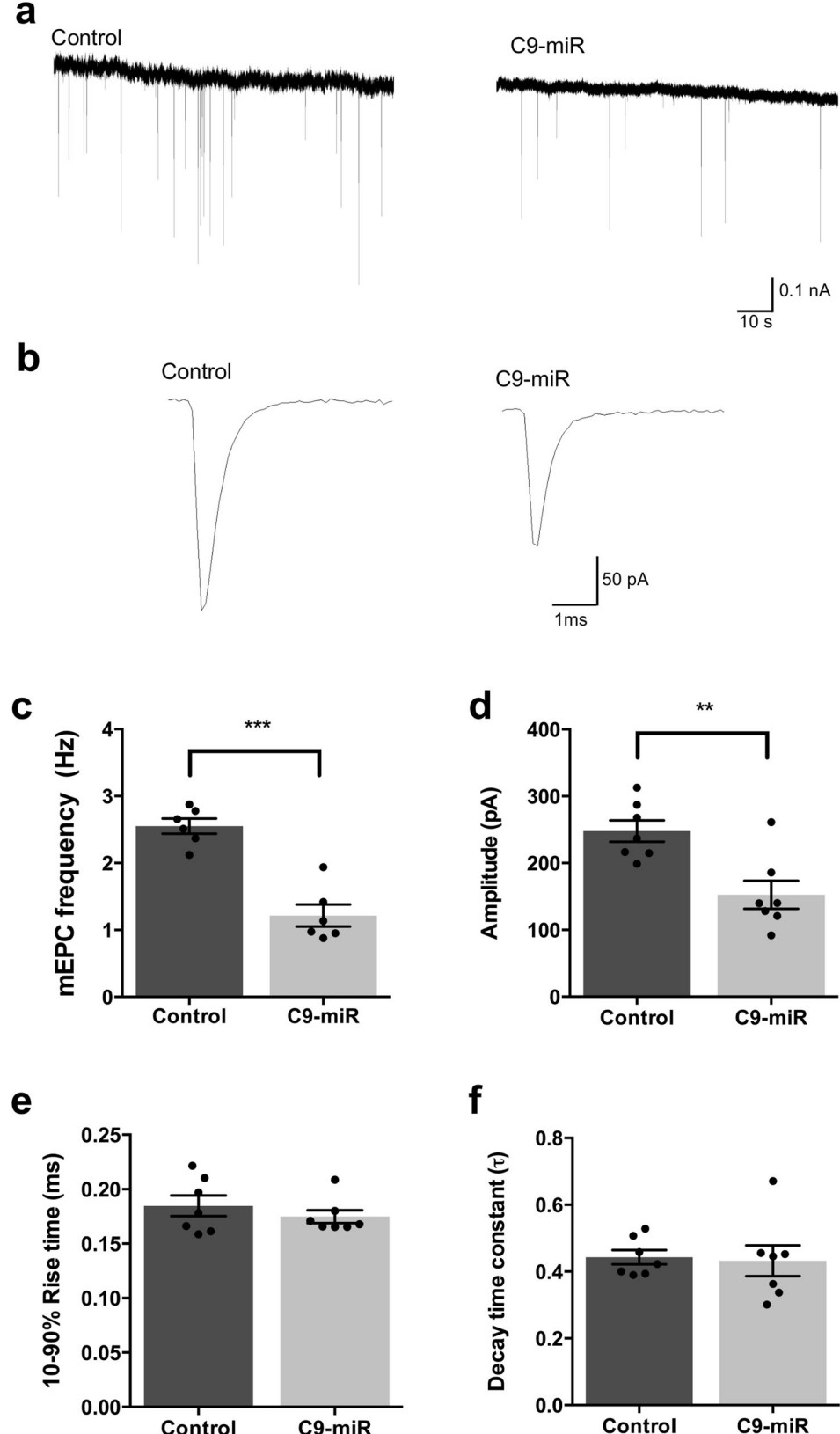

**Fig. 3 C9-miR zebrafish exhibited attenuated miniature endplate currents (mEPCs) at NMJs. a** Recordings of mEPCs, which result from spontaneous release of a quantum, were recorded in 6 dpf control and C9-miR fish ($n = 6$). **b** Representative mEPCs. Animals with reduced C9orf72 (C9-miR) displayed mEPCs with reduced frequency (**c**) ($n = 6$–7; ***$p < 0.0001$; Student's $t$ test) and amplitude (**d**) ($n = 7$; **$p < 0.001$; Student's $t$ test). Rise time (**e**) ($n = 7$; $p = 0.3947$; Student's $t$ test) and decay time (**f**) constant kinetics of mEPC ($n = 7$; $p = 0.8385$; Student's $t$ test) were not found to be significantly different between controls and C9-miR. Data are presented as mean ± SEM. $n$ represents the number of fish.

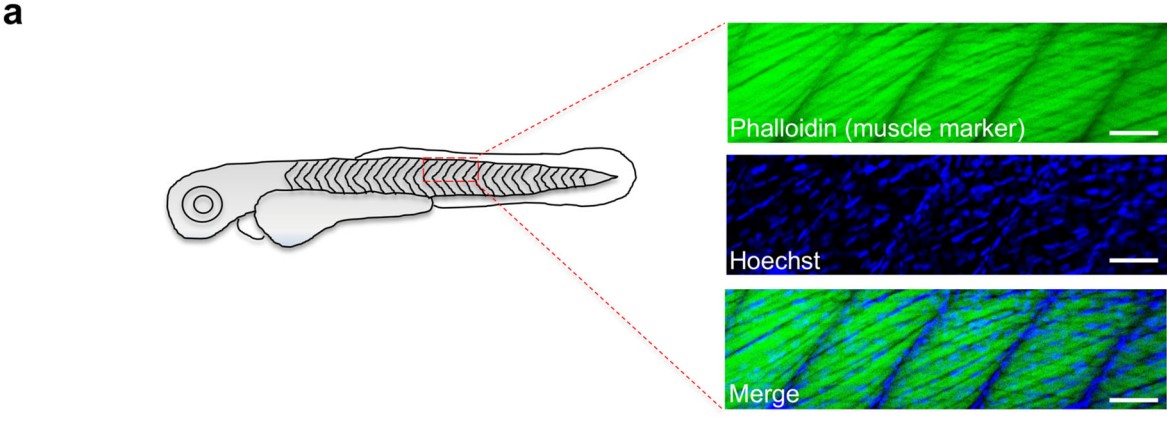

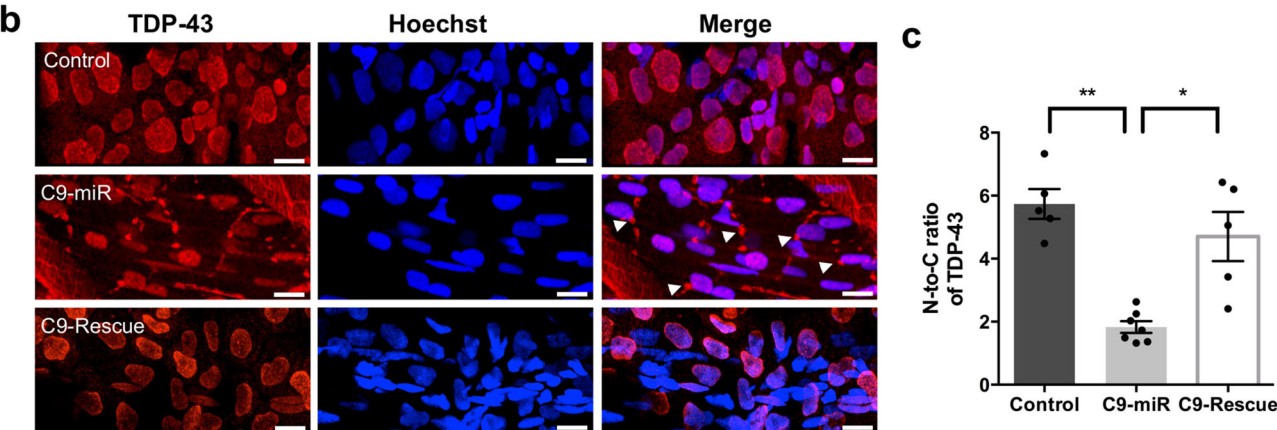

**Fig. 4 C9-miR zebrafish displayed TDP-43 pathology. a** Illustration of 6 dpf zebrafish skeletal muscle cells with large nuclei labelled with muscle marker, phalloidin, and nucleus marker, Hoechst. **b** Representative images of 6 dpf zebrafish skeletal muscle cells for TDP-43. Compared to control fish, we observed cytoplasmic clustering of TDP-43 expression in the C9-miR skeletal muscles that can be rescued in C9-rescue fish. Arrows indicate clusters of TDP-43 expression. **c** Quantification of the nucleus-to-cytoplasmic ratio for TDP-43. A significant reduction in N-to-C for TDP-43 was observed in C9-miR zebrafish ($n = 7$; **$p < 0.005$; one-way ANOVA) that can be significantly rescued upon the expression of the human *C9orf72* mRNA ($n = 5$; *$p < 0.05$; one-way ANOVA). Scale bar = 50 μm. Data are presented as mean ± SEM. *n* represents the number of fish.

SV2a was co-immunoprecipitated by RFP-C9orf72 but not by the RFP control (Fig. 7c). Taken together, these results suggest that C9orf72 interacts with SV2a.

Given that SV2a is an essential component of active zones and synaptic release machinery, we next sought to further assess synaptic activity at the NMJ by measuring SV cycling at the NMJ in zebrafish larvae using the fluorescent styryl dye, FM1-43[40,41]. C9-miR and control larvae were exposed to FM1-43 and its uptake into NMJ presynaptic boutons was monitored. The presynaptic terminals were acutely depolarized with a high [K+] Hank's Balanced Salt Solution (HBSS) (45 mM) to drive the exocytotic activity, SV cycle, load FM1-43 and label synaptic clusters. In controls, we observed strong fluorescence staining along terminal axon branches at individual synaptic varicosity boutons (Fig. 8a), while in C9-miR fish we found a significant reduction in FM1-43 loading in presynaptic terminals (Fig. 8b), indicating slowing of the exocytotic activity and the overall SV cycle. These findings reveal a key role for C9orf72 in regulating pre-SV release at NMJ.

To assess the organization of the presynaptic structure at the NMJ, we examined the expression of Rab3a, a protein associated with vesicles at active zones that is essential for SV release and neurotransmission (Fig. 8c–e). We found a reduced number of Rab3+ puncta in C9-miR fish compared to controls (Fig. 8c, d).

Additionally, the area of the putative synapses were smaller in C9-miR fish (Fig. 8e) compared to control fish.

## Discussion

Despite advances in studies of C9orf72-ALS, understanding the function of C9orf72 remains a key research element that is poorly explored. We generated a *C9orf72*-related ALS stable zebrafish line with a reduced expression of C9orf72. These fish display motor defects, muscle atrophy, motor neuron loss and mortality in early larval and adult stages. Additionally, they exhibit TDP-43 pathology, which is a key hallmark of ALS. Analysis of the structure and function of the NMJs revealed a significant reduction in the number of presynaptic and postsynaptic structures and an impaired release of quantal SVs at the NMJ in the C9-miR line. We also identified an important role of C9orf72 in controlling pre-SV trafficking and release at the zebrafish larval NMJ.

Reduced C9orf72 mRNA and protein levels have been reported in a range of patient tissues and patient-derived cell lines[1,6,7]. Our C9orf72 zebrafish model provides support to a LOF mechanism underlying *C9orf72*-dependent ALS. Our data are consistent with deletion or transient knockdown models in *Caenorhabditis elegans*[42] and zebrafish[43], respectively, showing defective motor phenotypes. However, in contrast, no motor neurons deficits were reported in *C9orf72* knockout mice[44–46]. The phenotypic discrepancy between *C9orf72* knockout mice and C9-miR fish might be explained by genetic compensation mechanisms that may occur upon complete C9orf72 deficiency (i.e. knockout) but not

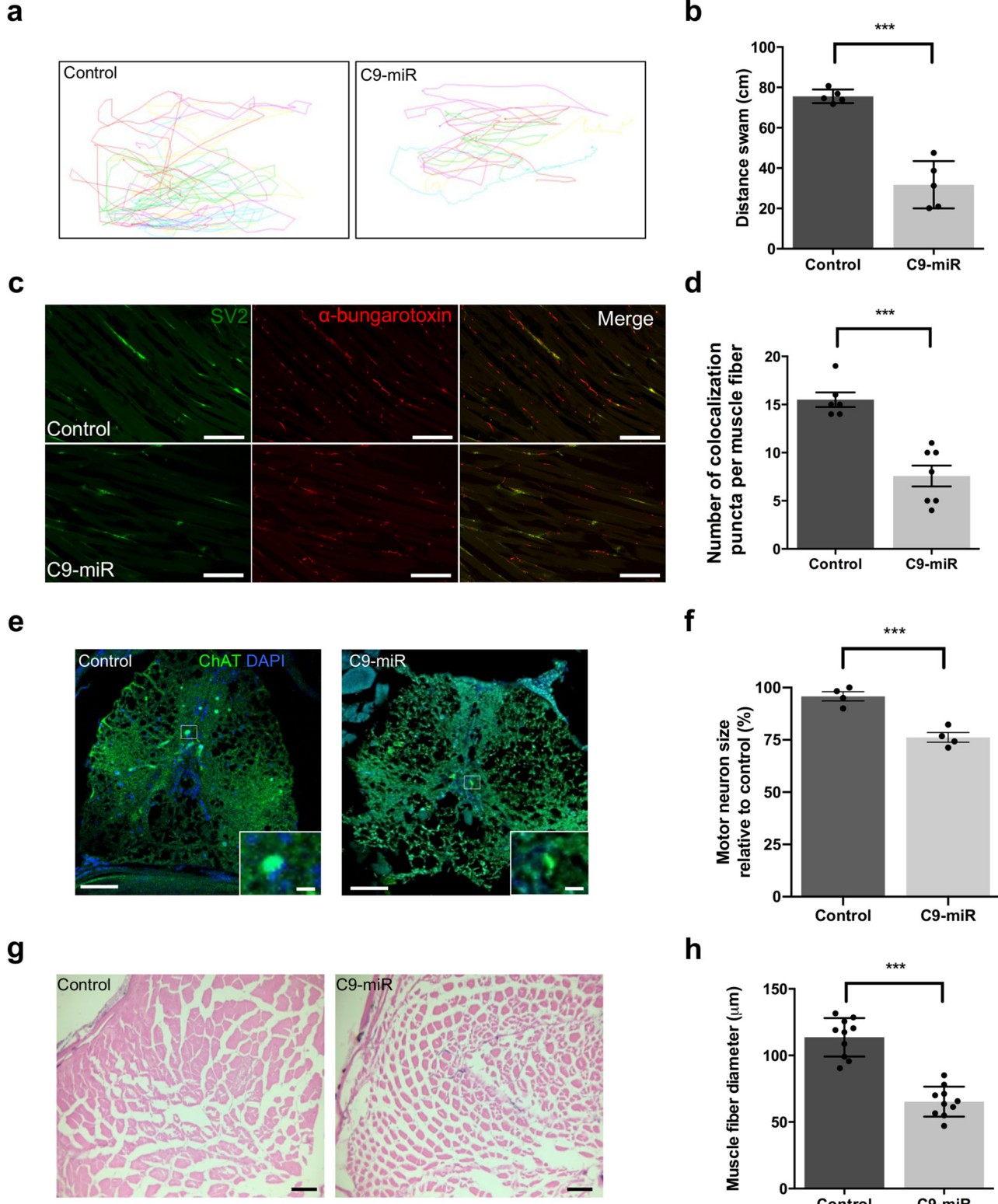

**Fig. 5 Adult zebrafish C9-miR display muscle atrophy, smaller motoneurons and behavioural deficits. a** Representative traces of swimming activity of five adult controls and C9-miR fishes (12-month old) during 30 s (left panel). **b** C9-miR fish exhibit behavioural deficits (right panel) ($n = 5$, ***$p < 0.0001$, Student's $t$ test). **c** Adult 12-month-old NMJs were examined in trunk section by co-immunostaining SV2 (green) and α-bungarotoxin (red).
**d** Quantification of the colocalizing presynaptic and postsynaptic clusters at NMJ in adult (12-month old) wild-type control ($n = 6$) and C9-miR fish ($n = 7$) (***$p < 0.0001$, Student's $t$ test). **e** ChAT staining in adult zebrafish spinal cord . **f** Large (mature) motor neurons (inset in **e**; scale bar = 10 μm) are reduced in size in C9-miR compared to controls ($n = 4$; ***$p < 0.0001$, Student's $t$ test). **g** Examination of adult zebrafish muscle myotomes by haematoxylin and eosin staining. **h** C9-miR fish display a smaller diameter of muscle fibres compared to controls ($n = 10$; ***$p < 0.0001$, Student's $t$ test). Data are presented as mean ± SEM. Scale bar = 50 μm. $n$ represents the number of sections from three adult fish per genotype.

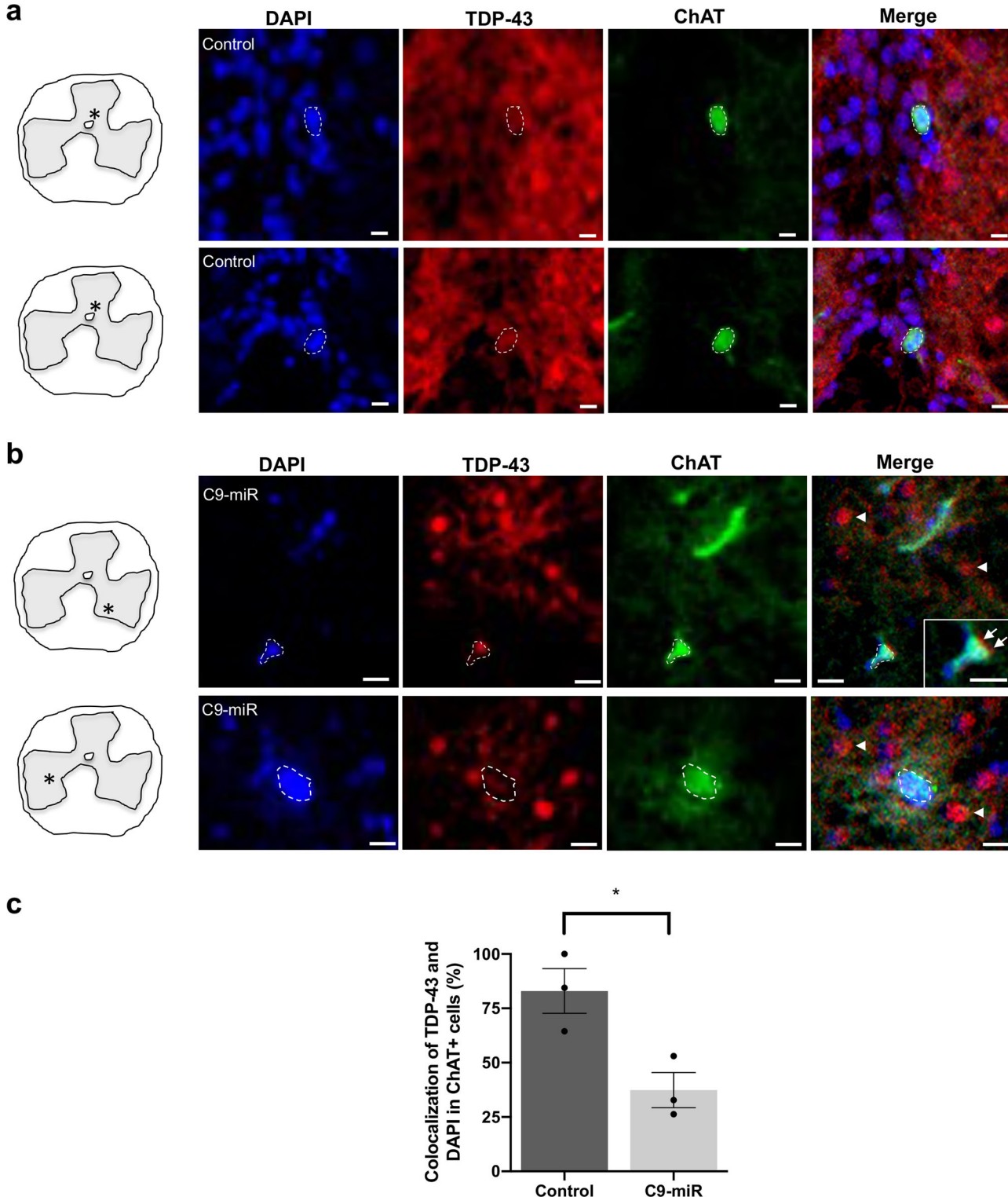

**Fig. 6 TDP-43 pathology in adult C9-miR zebrafish motor neurons.** Representative fluorescence images of spinal motor neurons immunostained with antibodies against ChAT (green), TDP-43 (red) and labelled with DAPI (blue) in 14–16-month-old adult (**a**) control (wild type) or (**b**) C9-miR spinal cord sections. Scale bar = 20 μm. Arrows and arrowheads illustrate TDP-43 mislocalization. **c** Quantification of colocalization (%) between the total cellular TDP-43 antibody signal and the nuclear DAPI stain in ChAT-positive motor neurons in C9-miR fish normalized to colocalization percentage in control fish ($n = 3$; *$p = 0.0251$, Student's $t$ test). Data are presented as mean ± SEM. $n$ represents the number of fish.

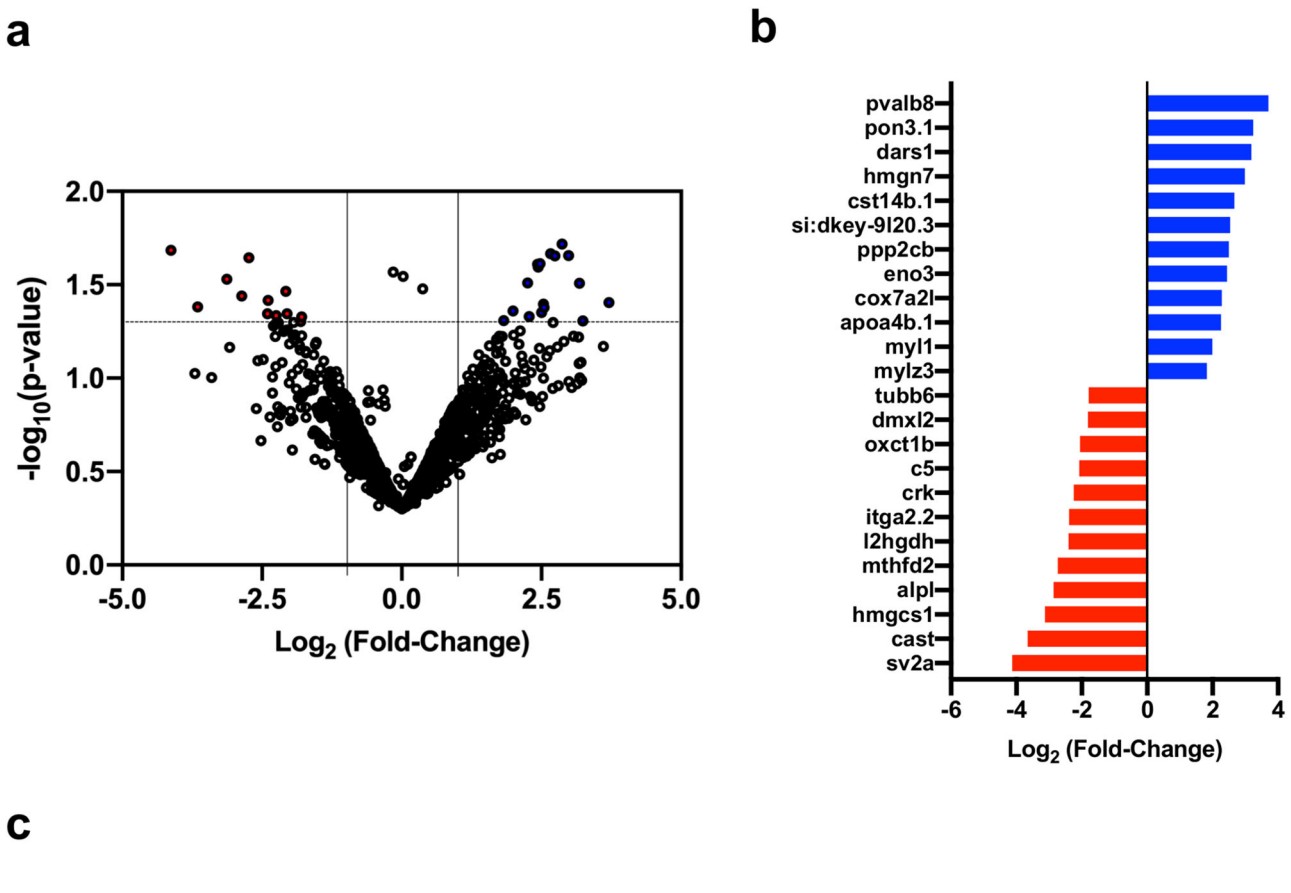

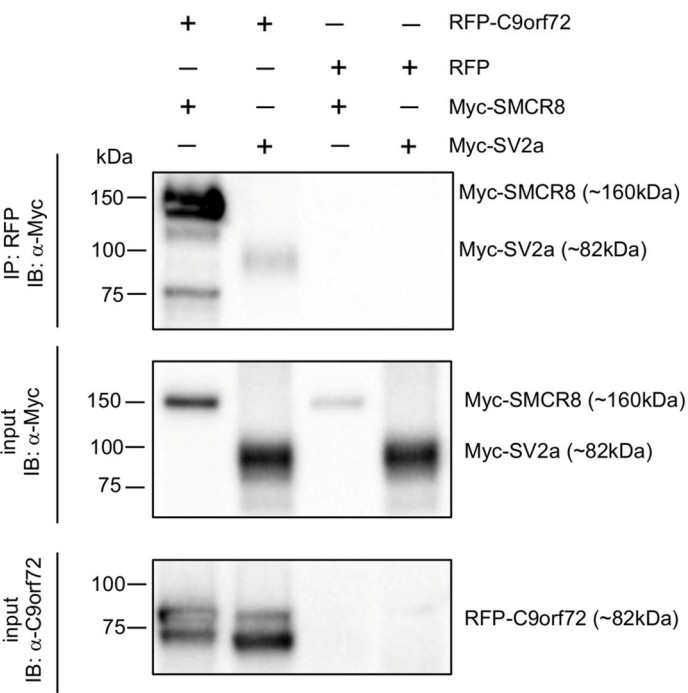

**Fig. 7 Sv2a interacts with C9orf72 and its protein expression is downregulated in C9-miR fish. a** Volcano plot showing the log2 fold change against the −log10 *p* value. Proteins in blue are upregulated while proteins in red are downregulated (log2 fold change of −1.5 and 1.5, delineated by vertical lines) and are significantly dysregulated (−log10 (*p* value) > 1.301, delineated by horizontal dotted line) between control and C9-miR fish (*N* = 4). **b** Differentially expressed proteins with *p* < 0.05 in C9-miR 6 dpf larvae (*N* = 4). **c** Immunoblot analysis of RFP-immunoprecipitated proteins and lysate of HEK293T cells co-expressing RFP-tagged C9orf72 or RFP in combination with Myc-tagged SV2a or Myc-tagged SMCR8 (a known interactor of C9orf72; positive control). Myc-SV2a and Myc-SMCR8 were extracted out of solution in samples where RFP-C9orf72 was co-transfected, but not in samples where RFP was co-transfected (*N* = 3).

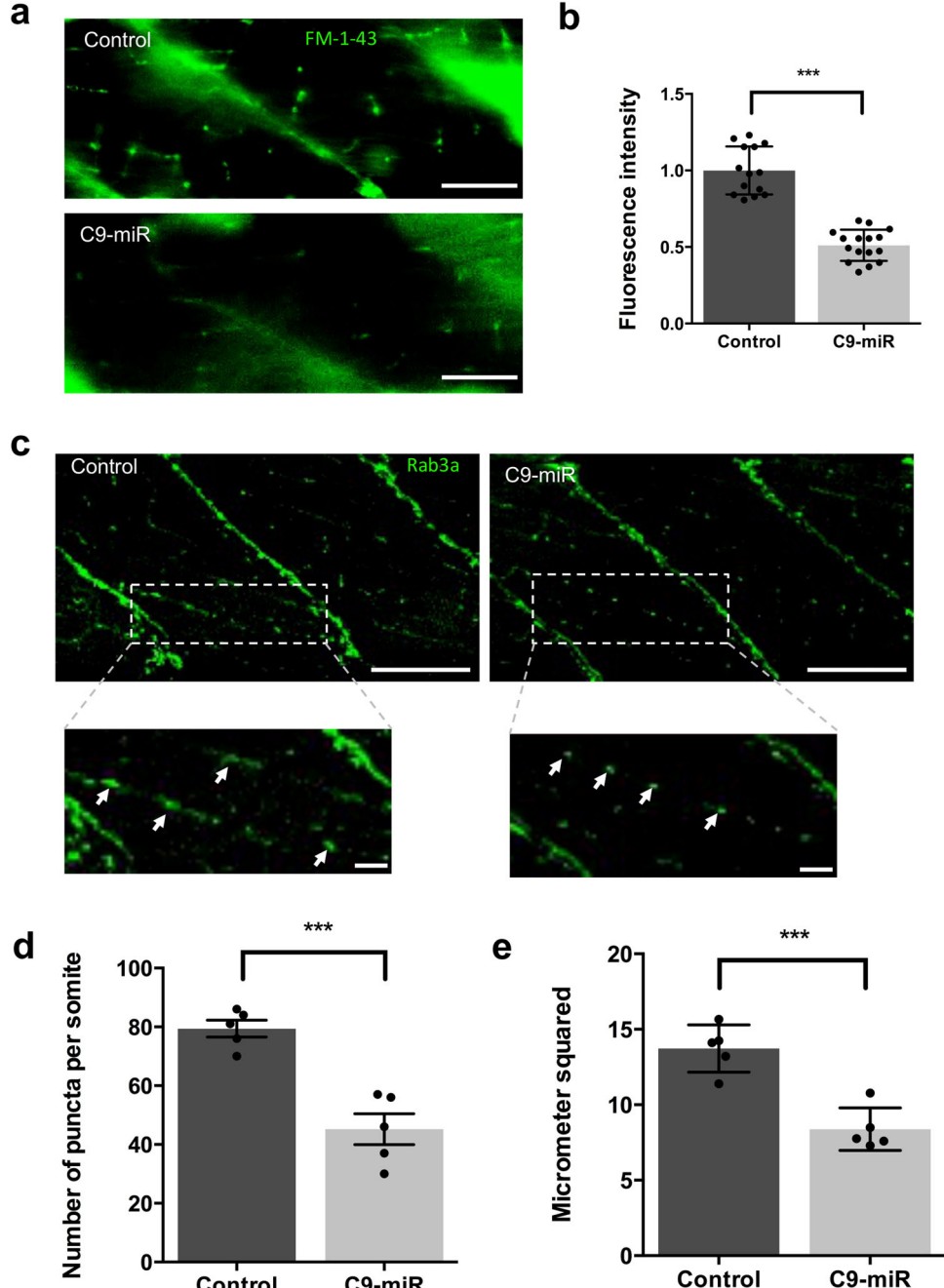

**Fig. 8 C9orf72 regulates synaptic active zones and activity at the neuromuscular junction. a** FM1-43 loading of NMJ boutons in 6 dpf fish. C9-miR fish displayed decreased FM1-43 loading compared to controls. **b** Quantification of FM1-43 fluorescent intensity in control ($n = 14$) and C9-miR fish ($n = 16$), showing a reduced FM1-43 fluorescent intensity in C9-miR fish ($p < 0.0001$; Student's $t$ test). **c** Putative synapses (arrows) were visualized with Rab3a immunostaining at 6 dpf (inset; scale bar = 10 μm). Rab3a+ synaptic puncta were reduced in number (**d**) and area (**e**) in 6 dpf C9-miR larvae when compared with wild-type controls ($n = 5$; ***$p < 0.0001$; Student's $t$ test). Scale bar = 50 μm. Data are presented as mean ± SEM. $n$ represents the number of fish.

upon partial loss of C9orf72 (i.e. knockdown). Importantly, complete deletion of *C9orf72* does not occur in C9orf72 ALS/FTD patients. In a recent elegant work, Shao and colleagues showed that C9orf72 protein dose reduction is critical for motor deficits in C9orf72 ALS/FTD mouse models[47], demonstrating the importance of C9orf72 haploinsufficiency in disease manifestation in mice rather than complete loss of C9orf72. Noteworthy, although differences exist in the homology of the human *C9orf72* orthologues in zebrafish and mice (i.e. 75% in zebrafish and 98% in mice), sequence differences likely have virtually little or no

bearing on the phenotypic discrepancy between a higher-order organism, *C9orf72* knockout mice and C9-miR fish, given that the motor deficits in C9-miR fish can be rescued by the expression of human *C9orf72* mRNA. Interestingly, in human motor neurons derived from normal individual iPSCs harbouring a CRISPR/Cas9-mediated *C9orf72* deletion[15] and in *C9orf72* knockout rats[48], rapid neurodegeneration and progressive motor deficits were, respectively, noted in response to excitotoxicity.

In addition, *C9orf72* knockout mice do not exhibit TDP-43 proteinopathy. The model presented here, importantly, displays

TDP-43 pathology and replicates haploinsufficiency as a major contributor to *C9orf72* ALS rather than being a full ablation of C9or72 LOF model. Intriguingly, the motor phenotypes observed in C9-miR zebrafish are consistent with several other zebrafish ALS models, including zebrafish model expressing C9orf72-related repeat expansions or DPR[27,49,50]. However, the presence of a reduced level of C9orf72 mRNA or protein in these models, as in ALS/FTD, was not examined in these studies. Of note, the expression of GGGGCC repeat expansions or DPR in zebrafish are toxic[49,51,52], consistent with several studies in neurons and other animals. We found that the expression of GGGGCC repeat expansions in our C9-miR fish exacerbated toxicity and resulted in death of zebrafish by 6 dpf (Supplementary Fig. 4). Such a synergistic interplay between reduced C9orf72 function and repeat-dependent gain of toxicity was observed in a recent study by Zhu and colleagues[53].

An important finding of this study is the synaptic impairments in C9-miR fish. The reduced frequencies and amplitudes of quantal neurotransmission events are consistent with observations made in several non-C9orf72 ALS models[21,22] and in tissue from patients with ALS[17]. We also report significant reductions in SV exocytosis and the number and area of putative synaptic puncta at NMJs. Additionally, we show a decrease in the expression of SV protein SV2a. These findings provide a novel role of C9orf72 in synaptic physiology at the presynaptic level. Interestingly, consistent with our findings, SV2a was also recently found at reduced levels in C9orf72-ALS patient-derived iPSC neurons[39]. Ablation of SV2a function in knockout models resulted in reduced number of readily releasable pool of SVs, diminished release probability and reduction in spontaneous synaptic events[54,55]. Intriguingly, similar observations of loss of SV2a and synaptic dysfunction were also observed in neurons expressing the C9orf72-related glycine-alanine (GA) DPR[39]. DPR proteins can disrupt pre-mRNA splicing in ALS/FTD patients[56]. It is possible that the expression of GA DPR in neurons reduces the level of *C9orf72* transcripts leading to the synaptic phenotypes. Additionally, we show that C9orf72 interacts with SV2a, suggesting that it may play a role in stabilizing the expression level of SV2a in presynaptic compartments.

Rab3a is important for transport of SVs and their docking at active zones[57]. It regulates synaptic transmission and it is associated with SVs through GEF activity[58,59]. For instance, at Rab3a-deficient terminals in mice, synaptic secretion response recovered slowly and incompletely following exhaustive stimulation[57]. In addition, the replenishment of docked vesicles following exhaustive stimulation at these terminals was also impaired[57]. Differentially expressed in normal and neoplastic domain containing proteins such as C9orf72 can function as Rab GEFs, enabling their activation, recruitment and interaction with downstream effectors[60,61]. A previous study had identified Rab3a as part of complex interacting with C9orf72[7]. Xiao et al. showed that Rab3a protein levels were unchanged in C9-WT and C9-knockout mice forebrain by western blot. Similarly, no significant change in Rab3a levels was observed in our proteomic analyses between controls and C9-miR fish. However, we observed reduced Rab3a-positive puncta at active zones at NMJs in C9-miR fish by immunostaining, suggesting that Rab3a-dependent SV transport to active zones may be impaired upon loss of C9orf72. It is, thus, plausible that in addition to the effect of reduced SV2a on synaptic dysfunction, SV exocytosis and quantal transmission defects in C9-miR fish may be exacerbated due to the altered function of C9orf72 as a GEF for Rab3a and, subsequently, its recruitment to SVs and role in SV transport.

In conclusion, we generated a stable C9orf72-LOF model in zebrafish that recapitulated some major hallmarks of ALS and enhanced our understanding of ALS pathogenesis. Importantly,

our findings demonstrate that loss of C9orf72 function impairs synaptic function at NMJs and result in motor deficits. We postulate that synaptic deficits observed in repeat expansions or DPR models may be the result of an indirect effect related to an impact of the repeats on C9orf72 levels.

## Methods

**Zebrafish husbandry.** Adult zebrafish (*Danio rerio*) were maintained at 28 °C at a light/dark cycle of 12/12 h in accordance with Westerfield zebrafish book[62]. Embryos were raised at 28.5 °C and collected and staged as previously described[63]. All the animal experiments were performed in compliance with the guidelines of the Canadian Council for Animal Care and received approval from the INRS-CNBE ethics committee.

**Anti-c9orf72 (synthetic miRNA) RNAi target site selection.** We first generated a template with c9orf72 RNA sequence, including 5'- and 3'-UTR sequence. 3'-UTR minimal sequence has been obtained from the analysis of data available on ensembl (http://asia.ensembl.org/) with zebrafish GRCz11 genome iteration and on Targetscan Fish website (http://www.targetscan.org/fish_62/). We analysed and annotated *c9orf72* sequence for identifying and avoiding selecting target sequence that would run across (i) potential polymorphisms in the 3'-UTR sequence and (ii) endogeneous miRNA. Based on these data, we selected 4× unique target sites on the 3'-UTR sequence of *c9orf72* that do not show any off-specific match across the zebrafish genome. Each site and corresponding mature anti-c9orf72 synthetic miRNA are presented in Supplementary Table 1.

**RNAi plasmid generation.** To generate the *c9orf72*-RNAi transgene (Tol2-UBI:dsRED:*c9orf72*-1234-Cryst:eGFP) used in this study to silence the gene *c9orf72*, we first used a previously generated empty RNAi-plasmid compatible with the tol2-kit, pME-RNAi642[24]. Based on this design and following previous instructions, we designed 4× anti-*c9orf072* miRNAs stem loops compatible with the pME-RNAi642 (Supplementary Table 1). pME-RNAi642 was digested with BsmBI and gel-extracted. Each stem loops (4×) were annealed and inserted into pME-RNAi642 following previous instructions[24]. In all, 4× different pME-RNAi-*c9orf72* has been generated and named pME-RNAi-*c9orf72*-1 to -4. We further chained the 4× stem loop. We ended with a 4×-anti-*c9orf72* RNAi pME plasmid named pME-RNAi-*c9orf72*-1234. In parallel, a custom-made 1456-pDEST-miniTol2-R4-R2_Cryst:eGFP clone was generated; this clone presents miniTol2 sequence surrounding gateway Att-R4 and Att-R2 sequences followed by a Cryst:eGFP cassette (Crystallin-promoter driving eGFP into lenses for identifying transgenic/carrier fish). Following the manufacturer's instruction, we performed a Gateway LR-reaction mixing/recombining p5E-Ubi (Ubiquitin promoter) and pME-RNAi-*c9orf72*-1234 into 1456-pDEST-miniTol2-R4-R2_Cryst:eGFP. The final plasmid obtained was named Tol2-UBI:dsRED:*c9rof72*-1234-Cryst:eGFP and was used to perform one-cell-stage injections for transgene integration.

**Injections for transgene integration and rescue experiments.** To integrate Tol2-UBI:dsRED:*c9orf72*-1234-Cryst:eGFP construct into the zebrafish genome, 1 nL of a mix of 30 ng/μL of construct and 25 ng/μL of Transposase mRNA was injected into one-cell-stage embryos using the Picospritzer III pressure ejector. Rescue experiment was performed using human *C9orf72* mRNA (NM_001256054.3) that was produced via an open-reading frame clone of human *C9orf72* long transcript purchased from GeneCopoeia. In vitro transcription was done using T7 Message Machine Kit (Ambion) and 1 nL of *C9orf72* mRNA (100 ng/μL) was injected into the one-cell-stage embryos.

**Western blot.** Larvae were collected from 6 dpf fish and for each condition 30 larvae were lysed in 150 μL lysis buffer containing 150 mM NaCl, 50 mM Tris-HCl pH 7.5, 1% triton, 0.1% sodium dodecyl sulfate (SDS), 1% sodium deoxycolate and protease inhibitor cocktail (1:10, Sigma-Aldrich). The lysates were then centrifuged at 10,000 rpm for 10 min at 4 °C. The supernatant was collected and protein concentration was estimated using Bradford assay (BioRad). Western blotting was performed using 40 μg lysate per sample, which were resolved on a 7.5% SDS-polyacrylamide gel (BioRad). After electrophoresis, proteins on the gel were electrotransferred onto polyvinylidene difluoride (PVDF) mini-membranes (BioRad). The membranes were blocked with 5% non-fat milk solution in 1× phosphate-buffered saline (PBS) or with 5% bovine serum albumin (BSA; Sigma) in 1× Tris buffered saline for immunoblotting with Novus Npb2-15656 (1:5000) antibody against C9orf72 and Sigma A5441 (1:5000) antibody used against β-actin. Detection was performed using goat anti-rabbit and goat anti-mouse antibodies, respectively, conjugated with horseradish peroxidase (HRP). Bands were visualized with ECL and imaged using ChemiDoc (BioRad). Quantifications were performed with Image Lab (BioRad), normalizing using β-actin.

**Gene expression study.** RNA was isolated from ~30 embryos using TriReagent® (Sigma) according to manufacturer's protocol. In all, 1 μg of RNA was used for cDNA synthesis by the SuperScript®Vilo™ Kit (Invitrogen). RT-qPCR was run with

SYBR Green Master Mix (Bioline) using the LightCycler® 96 (Roche). *ef1a* was used as the reference gene for normalization and following primers were used for *C9orf72*: FW: 5'-GTGTGCCAGAGGAGGTTGAT-3'; RV:5'-ACAGCTGTCTCC AATATCATCG-3'.

**Gross morphology and survival assessment**. Larvae were assessed for their survival rate and morphological phenotypes. The sample sizes for control and C9-miR were as follows: three different batches ($N = 3$) each batch containing 25 larvae ($n = 25$) for both control and C9-miR. Gross morphology was observed under a stereomicroscope (Leica S6E).

**GGGGCC expansion repeat microinjections**. GGGGCC repeat constructs (p3s and p91s) were kindly provided by Dr. Ludo Van Den Bosch and Dr. Adrian Isaacs. Synthesis of mRNAs and microinjections were performed as previously described[50].

**Behavioural assay**. Larvae (6 dpf) were separated into single wells of a 96-well plate containing 200 μL of E3 media and habituated in the Daniovision® recording chamber (Noldus) for 1 h before start of experiment. Larval locomotor activity was monitored over light–dark cycles using the Daniovision® apparatus. Analysis was performed using the Ethovision XT12 software (Noldus) to quantify the total swimming distance in given hours and the locomotor activity per second.

**NMJ morphology in larval zebrafish**. Immunohistochemical analyses were performed on 2 and 6 dpf zebrafish to visualize NMJ presynaptic and postsynaptic structures. Briefly, animals were fixed in 4% paraformaldehyde overnight at 4 °C. After fixation, the larvae were rinsed several times (1 h) with PBS-Tween and then incubated in PBS containing 1 mg/mL collagenase (30 min for 2 dpf fish and 180 min for 6 dpf fish) to remove skin. The collagenase was washed off with PBS-Tween (1 h), and the larvae were incubated in blocking solution (2% normal goat serum (NGS), 1% BSA, 1% dimethyl sulfoxide (DMSO), 1% Triton-X in PBS) containing 10 mg/mL tetramethylrhodamine-conjugated α-bungarotoxin (Ther-mofisher T1175) for 30 min. The larvae were rinsed several times with PBST (30 min) and then incubated in freshly prepared block solution containing primary antibody SV2 (1:200, Developmental Studies Hybridoma Bank) overnight at 4 °C. Following this, larvae were incubated in block solution containing a secondary antibody (Alexa fluor 488, 1:1000, cat# A-21042, Invitrogen) overnight at 4 °C. The following day, the larvae were washed several times with PBST and mounted on a glass slide in 70% glycerol. Slides were blinded for Z-stack imaging with a Zeiss LSM780 confocal microscope (Carl Zeiss, Germany). The images were then processed with the ZEN software (Carl Zeiss). Colocalization of presynaptic and postsynaptic structures per somite were counted per fish from a set of stacked Z-series images using ImageJ (NIH).

**TDP-43 immunostaining**. Larval zebrafish (6 dpf) were fixed in 4% paraformaldehyde overnight at 4 °C. After fixation, the larvae were rinsed several times (1 h) with PBS-Tween and then incubated in PBS containing 1 mg/mL collagenase (30 min for 2 dpf fish and 180 min for 6 dpf fish) to remove skin. The collagenase was washed off with PBS-Tween (PBST) (1 h), and the larvae were incubated in blocking solution (2% NGS, 1% BSA, 1% DMSO, 1% Triton-X in PBS) for 30 min. The larvae were rinsed several times with PBS-Tween (30 min) and then incubated in block solution containing primary antibody TDP-43 (1:200, Sigma Aldrich) overnight at 4 °C. Following this, larvae were incubated in block solution containing a secondary antibody (Alexa fluor 555, 1:1000, Invitrogen) overnight at 4 °C. The following day, the larvae were washed several times with PBST and incubated in Hoechst solution for 15 min. Larvae were then washed in PBST and mounted in glycerol. Z-stack images were taken using a Zeiss LSM780 confocal microscope (Carl Zeiss, Germany). The images were then processed with the ZEN software (Carl Zeiss). Images were acquired using the same settings. Researcher was blinded to genotype prior to image analysis. Each image was processed using the Fiji (ImageJ) software, and analysis was performed in the skeletal muscles in 50 μm-by-50 μm areas. A threshold was used to define the nuclei area and cytoplasm was defined as outside the nuclei. ImageJ Plugin versatile ward tool was used to detect nucleus and the remaining large area devoid of nuclei was considered as the cytoplasm. TDP-43 mean intensity was measured in these two areas. The ratios were then analysed in GraphPad Prism6 and Kruskal–Wallis test was performed to calculate the statistical significance.

**H&E staining**. For H&E staining, spinal cord sections (15 μm) sections of adult fish were stained with haematoxylin (Statlab) for 4 min and washed with alcohol-acid and were rinsed with tap water. The sections were then soaked in saturated lithium carbonate solution for 10 s and then rinsed with tap water. Finally, staining was performed with Eosin Y (Statlab) for 2 min and mounted under coverslip with permount mounting media.

**NMJ staining on adults**. Adult wild-type and C9-miR zebrafish were euthanized and fixed in 4% paraformaldehyde for 72 h at 4 °C. The fish were then embedded in paraffin and longitudinal sections (15 μm) were obtained on a Leica microtome and processed for immunostaining. Slides with the sections were incubated in xylene twice, followed by rehydration in 4 successive baths of 100, 95, 70 and 50% ethanol in distilled water, respectively. Sections were blocked for 30 min with blocking solution (2% NGS, 1% BSA, 1% DMSO, 1% Triton-X in PBS) containing 10 mg/mL tetramethylrhodamine-conjugated α-bungarotoxin (Thermofisher T1175) to label postsynaptic acetylcholine receptors. Sections were then rinsed several times with PBST and then incubated with primary antibody SV2 (1:200, Developmental Studies Hybridoma Bank) overnight at 4 °C. After washing, sections were incubated in block solution containing a secondary antibody (Alexa fluor 488, 1:1000, cat# A-21042, Invitrogen) for 2 h, followed by rising and mounting. Slides were blinded for Z-stack imaging with a Zeiss confocal microscope, and the images were then processed with the ZEN software (Carl Zeiss). Differences in NMJ integrity were determined by examining SV2 and αBTX colocalization per muscle fibre on each slice section per fish to calculate the number of innervated junctions using ImageJ.

**Motoneuron staining and TDP-43 pathology in adult fish**. Adult wild-type and C9-miR zebrafish were euthanized and fixed in 4% paraformaldehyde for 72 h at 4 °C. The fish body trunk was cross-sectioned using a microtome at 15-μm-thick slices. Sections were incubated in xylene twice, followed by rehydration in 4 successive baths of 100, 95, 70 and 50% ethanol in distilled water, respectively. They were then rinsed several times in PBS and incubated in citrate buffer (1 M, pH 6) for antigen retrieval. After several washes in PBST, sections were incubated in blocking buffer (1% NGS, 0.4% Triton-X) for 1 h, followed by incubation in primary antibody ChAT (1:500; Invitrogen) alone or with primary antibody TDP-43 (1:200; Sigma Aldrich) at 4 °C overnight. The next day, sections were washed and incubated in blocking solution containing secondary antibody Alexa Fluor 488 (1:750; Molecular Probes, Invitrogen) for 2 h, followed by rinsing and mounting in Prolong Gold antifade reagent with DAPI (Invitrogen). Motor neurons in spinal cord were identified as ChAT-positive objects >10 μm per section[64]. Researcher was blinded to genotype during quantification and analysis. Motor neurons in spinal cord cross-sections were quantified using Fiji ImageJ (NIH). For assessment of TDP-43 pathology, the ROI manager tool in Fiji ImageJ was used to define ChAT+ motor neurons in TDP-43-stained sections and the colocalization between the cellular TDP-43 antibody signal and the nuclear DAPI stain in these neurons were measured using Fiji ImageJ and analysed.

**FM1-43 staining**. Zebrafish larvae (6 dpf) were first anaesthetised in Evan's solution (134 mM NaCl, 2.9 mM KCl, 2.1 mM CaCl₂, 1.2 MgCl₂, 10 mM HEPES, 10 mM glucose) containing 0.02% tricaine (Sigma Aldrich). The larvae were then pinned to a Sylgard-coated dish both at the head and extreme tail end using electrolytically sharpened tungsten needles. The skin was then carefully peeled away to expose the muscles and to permit access to FM1-43 (Molecular Probes). The fish were treated with Evan's solution containing 10 μM of FM1-43 to allow preloading penetration of the dye molecules. After 10 min, the fish were transferred to a high potassium HBSS (97 mM NaCl, 45 mM KCl, 1 mM MgSO₄, 5 mM HEPES, 5 mM CaCl₂) containing 10 μM of FM1-43 for 5 min. The fish were then incubated with an Evan's solution with 10 μM of FM1-43 finished for an additional 3 min, after which loading was complete. The larvae were then washed with a low calcium Evan's solution (0.5 mM CaCl₂) three times for 5 min to minimize spontaneous release of loaded SVs. The fish were imaged for FM1-43 staining using a ×40 Examiner A1 microscope (Zeiss). Blind measurements of FM1-43 staining at NMJs in wild-type control and C9-miR fish were performed per three somites per fish for each genotype using Fiji ImageJ (NIH).

**Electrophysiology recordings**. Whole-cell patch clamp recordings were taken from muscle cells of 6 dpf larvae. The preparation was bathed in an extracellular solution (134 mM NaCl, 2.9 mM KCl, 1.2 mM MgCl, 10 mM HEPES, 10 mM glucose, pH 7.8) containing 1 μM of tetrodotoxin (Tocris, UK) in order to block action potentials during mEPC recordings. Patch clamp electrodes (2–4 MΩ) were filled with an intracellular solution (130 mM CsCl, 8 mM NaCl, 2 mM CaCl₂, 10 mM HEPES, 10 mM EGTA, 4 Mg-ATP, 0.4 Li-GTP, pH 7.4). mEPCs from white muscle fibres were recorded in the whole-cell configuration with an Axopatch 200B amplifier (Molecular Devices) at a holding potential of −60 mV, low-pass filtered at 5 kHz and digitized at 50 kHz. Series resistance was compensated by at least 85% using the amplifier's compensation circuitry. Synaptic currents were acquired using the pCLAMP10 software (Molecular Devices).

**Analysis of mEPCs**. mEPCs were analysed using the AxoGraph X software. The mEPC recordings were examined by the software, and synaptic events were detected using a template function. Overlapping or misshapen events were removed and the remaining events were averaged and the properties (amplitudes, decay time constants and frequencies) of the averaged trace were measured. Events with slow rise times and low amplitudes were excluded from the analysis, therefore only fast rise time events were included in our analysis since these events originated from the cells that were patch clamped rather than from nearby, electrically coupled muscles. Single decay time constants were fit over the initial (fast) decay portion and over the distal (slow) portion of the decay. For each *n*, currents were recorded from a single white muscle fibre from a single larva. Zebrafish twitch white fibres make up the bulk of the trunk musculature, and they are easily

identifiable under the microscope. We focussed on white fibres as mammalian skeletal musculature is mostly comprised of twitch fibre types.

**Rab3a immunostaining in larval zebrafish.** Larval zebrafish (6 dpf) were fixed in 4% paraformaldehyde overnight at 4 °C. After fixation, the larvae were rinsed several times (1 h) with PBS-Tween and then incubated in PBS containing 1 mg/mL collagenase (30 min for 2 dpf fish and 180 min for 6 dpf fish) to remove skin. The collagenase was washed off with PBS-Tween (1 h), and the larvae were incubated in blocking solution (2% NGS, 1% BSA, 1% DMSO, 1% Triton-X in PBS) for 30 min. The larvae were rinsed several times with PBST (30 min) and then incubated in freshly prepared block solution containing primary antibody Rab3a (1:100, Sigma Aldrich, cat# WH0005864M1) overnight at 4 °C. Following this, larvae were incubated in block solution containing a secondary antibody (Alexa fluor 488, 1:1000, Invitrogen) overnight at 4 °C. The following day, the larvae were washed several times with PBST and mounted on a glass slide in 70% glycerol. Z-stack images were taken using a Zeiss LSM780 confocal microscope (Carl Zeiss, Germany). The images were then processed with the ZEN software (Carl Zeiss). Researcher was blinded to genotype prior to image analysis. The number and area of Rab3a-positive puncta per three somites were counted and measured per fish using Fiji ImageJ (NIH).

**Mass spectrometry (MS) sample preparation.** Proteins were extracted with the protocol used for western blot protein extraction. Then a 1:8:1 ratio was used to precipitate proteins, 1× of cell lysates, 8× of 100% ice-cold acetone and 1× of 100% trichloroacetic acid in low binding protein tubes. In all, 20 μg of proteins were precipitated. Proteins were incubated at −20 °C for 12 h and centrifuged at 11,500 rpm for 15 min at 4 °C. Supernatant was then discarded.

A standard TCA protein precipitation was first performed to remove detergents from the samples (or acetone precipitation). Protein extracts were then re-solubilized in 10 μL of a 6 M urea buffer. Proteins were reduced by adding 2.5 μL of the reduction buffer (45 mM dithiothreitol, 100 mM ammonium bicarbonate) for 30 min at 37 °C and then alkylated by adding 2.5 μL of the alkylation buffer (100 mM iodoacetamide, 100 mM ammonium bicarbonate) for 20 min at 24 °C in dark. Prior to trypsin digestion, 20 μL of water was added to reduce the urea concentration to 2 M. In all, 10 μL of the trypsin solution (5 ng/μL of trypsin sequencing grade from Promega, 50 mM ammonium bicarbonate) was added to each sample. Protein digestion was performed at 37 °C for 18 h and stopped with 5 μL of 5% formic acid. Protein digests were dried down in a vacuum centrifuge and stored at −20 °C until liquid chromatography tandem mass spectrometry (LC-MS/MS) analysis.

**Mass spectrometry (LC-MS/MS).** Prior to LC-MS/MS, protein digests were re-solubilized under agitation for 15 min in 10 μL of 0.2% formic acid. Desalting/cleanup of the digests was performed by using C18 ZipTip pipette tips (Millipore, Billerica, MA). Eluates were dried down in a vacuum centrifuge and then re-solubilized under agitation for 15 min in 12 μL of 2% acetonitrile (ACN)/1% formic acid. The LC column was a PicoFrit fused silica capillary column (15 cm × 75 μm i.d.; New Objective, Woburn, MA), self-packed with C-18 reverse-phase material (Jupiter 5 μm particles, 300 Å pore size; Phenomenex, Torrance, CA) using a high-pressure packing cell. This column was installed on the Easy-nLC II system (Proxeon Biosystems, Odense, Denmark) and coupled to the Q Exactive (ThermoFisher Scientific, Bremen, Germany) equipped with a Proxeon nanoelectrospray Flex ion source. The buffers used for chromatography were 0.2% formic acid (buffer A) and 100% ACN/0.2% formic acid (buffer B). Peptides were loaded on-column at a flow rate of 600 nL/min and eluted with a 2 slope gradient at a flow rate of 250 nL/min. Solvent B first increased from 2 to 40% in 120 min and then from 40 to 80% B in 20 min. LC-MS/MS data were acquired using a data-dependent top17 method combined with a dynamic exclusion window of 7 s. The mass resolution for full MS scan was set to 60,000 (at $m/z$ 400) and lock masses were used to improve mass accuracy. The mass range was from 360 to 2000 $m/z$ for MS scanning with a target value at 1e6, the maximum ion fill time at 100 ms, the intensity threshold at 1.0e4 and the underfill ratio at 0.5%. The data-dependent MS2 scan events were acquired at a resolution of 17,500 with the maximum ion fill time at 50 ms and the target value at 1e5. The normalized collision energy used was 27 and the capillary temperature was 250 °C. Nanospray and S-lens voltages were set to 1.3–1.7 kV and 50 V, respectively.

The peak list files were generated with Proteome Discoverer (version 2.3) using the following parameters: minimum mass set to 500 Da, maximum mass set to 6000 Da, no grouping of MS/MS spectra, precursor charge set to auto, and minimum number of fragment ions set to 5. Protein database searching was performed with Mascot 2.6 (Matrix Science) against the Refseq Danio Rerio protein database. The mass tolerances for precursor and fragment ions were set to 10 ppm and 0.1 Da, respectively. Trypsin was used as the enzyme allowing for up to one missed cleavage. Cysteine carbamidomethylation was specified as a fixed modification and methionine oxidation as variable modifications. Data analysis was performed using Scaffold (version 4.8).

**Plasmid constructions.** Human *C9orf72* cDNA (GeneCopoeia) was cloned into pmRFP-C1 plasmid (Addgene, #54764) to generate RFP-C9orf72 plasmids. Myc-SMCR8 and Myc-SV2a plasmids were purchased from GeneCopoeia.

**Cell transfection, immunoprecipitation and western blotting.** HEK293T cells were co-transfected with 4 μg of RFP-C9orf72 or RFP (as negative control) and 4 μg of Myc-SMCR8 or Myc-SV2a plasmids using polyethylenimine. The cells were rinsed 3× with cold PBS before being lysed with 1 mL of RIPA buffer (0.01 M PBS, 0.0027 M KCl, 0.137 M NaCl, 0.1%$_{v/v}$ SDS, 0.5%$_{m/v}$ Na-deoxycholate, 1% NP-40, pH 7.4) with protease inhibitors for 15 min on ice. The RIPA was then collected into micro-centrifuge tubes and centrifuged at 21,000 × $g$ for 15 min at 4 °C. The supernatant was then split into input and RFP-Trap fractions. In all, 50 μL of lysate was mixed with 50 μL of 2× Laemmli buffer and stored at −20 °C. The remaining lysate was mixed with 20 μL of RFP-Trap agarose beads (twice rinsed with 500 μL RIPA buffer) and incubated for 2 h. The beads were then rinsed 5 times with RIPA buffer before the samples were resuspended in 60 μL of 1× Laemmli buffer. The samples with SMCR8 were then denatured at 94 °C for 5 min while samples with the transmembrane SV2a partner were left at room temperature.

Twenty microlitres of the samples were then loaded into a 7.5% gel for SDS-polyacrylamide gel electrophoresis at 150 V for 1 h, before being transferred to a PVDF membrane for 30 min in a BioRad Turboblot. The membranes were then placed in blocking solution (5% milk powder in PBS-T) before being incubated overnight with rabbit anti-C9orf72 (1:1000; Abcam #ab221137) or mouse anti-Myc antibodies (1:1000; Sigma #M5546). The membranes were then rinsed 3 times with 10 mL PBS-T before being incubated at room temperature in 10 mL of HRP-conjugated secondary antibodies (HRP goat anti-rabbit or HRP goat anti-mouse; Sigma). The membranes were then rinsed 3 times in 10 mL PBS-T before ECL solution (BioRad) was added to the membranes for imaging on a BioRad imager. The imaged blots were quantified using ImageLab.

**Statistics and reproducibility.** All zebrafish experiments were performed on at least three replicates ($N$) and each consisted of a sample size ($n$) of 5–65 fish. Data are presented as mean ± SEM. Significance was determined using either Student's $t$ test or one-way analysis of variance (ANOVA) followed by multiple comparisons test. Tukey post hoc multiple comparisons test was used for normally distributed and equal variance data. Kruskal–Wallis ANOVA and Dunn's method of comparison were used for non-normal distributions. All graphs were plotted using the Graphpad PRISM software. Significance is indicated as *$p < 0.05$, **$p < 0.001$ and ***$p < 0.0001$.

**Reporting summary.** Further information on research design is available in the Nature Research Reporting Summary linked to this article.

## Data availability

Source data underlying the figures are available in Supplementary Data 1. The mass spectrometry proteomics data have been deposited to the ProteomeXchange Consortium via the PRIDE partner repository with the data set identifier PXD025850 and doi: 10.6019/PXD025850. The data represented in this manuscript as well as the material used in this study will be available on reasonable request.

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

## Acknowledgements

This study was supported by the Society of ALS Canada and Brain Canada (to S.A.P.). S.A.P. is supported by the Natural Science and Engineering Research Council, Canadian Foundation for Innovation (CFI), Canadian Institutes of Health Research (CIHR), an ALS Canada-Brain Canada Career Transition Award and a FRQS Junior 1 research scholar. S.A.P. holds the Anna Sforza Djoukhadjian research chair in ALS. J.G. is

supported by a NHMRC Investigator Grant (Fellowship APP1174145), a NHMRC Project Grant (APP1165850) and a Rebecca L. Cooper Medical Research Project Grant (PG2019405). The authors would like to thank summer student Gabrielle Fortier for her assistance in some of the zebrafish experiments.

## Author contributions

S.A.P. conceived this work. Z.B. designed, collected, analysed and interpreted the results from studies related to the characterization of the C9-miR zebrafish line and synaptic defects. Y.E.P. collected and analysed the results of Co-IP experiments. J.G. and S.A.P. generated the C9-miR zebrafish line. S.A.P. and Z.B. performed the electrophysiological analyses. Z.B., J.G., Y.E.P. and S.A.P. interpreted the results. S.A.P. secured the research funding. Z.B., J.G. and S.A.P. drafted the manuscript. Z.B. and S.A.P. with contributions from all authors prepared the final version of the manuscript. All authors read the final version of this manuscript.

## Competing interests

The authors declare no competing interests.
