## [Transparent Peer Review File · Communications Biology]

Reviewers' comments:

Reviewer #1 (Remarks to the Author):

Summary

In this paper Butti et al. describe the generation and characterisation of transgenic zebrafish line whereby miRNA is used to knock-down the C9orf72 gene. The authors show that reduced C9orf72 function leads to motor defects, muscle atrophy, motor neuron loss and premature death. They then analyse the structure and functional capacity of the neuromuscular junctions (NMJs) and found that the C9-miRNA zebrafish have reduced functional NMJs from 6 dpf. Finally, the authors perform proteomic analysis and find a downregulation of synaptic proteins in the C9-miRNA fish. Investigation of synaptic vesicle cycling at the NMJ indicates C9-miRNA fish have slowed exocytotic activity of synaptic vesicles and conclude that C9orf72 plays a key role in regulating pre-synaptic vesicle release at the NMJ.

While this is an interesting finding, there is lack of novelty as several other prior studies have already concluded that C9orf72 protein plays a role in vesicle cycling. In addition, numerous concerns with the experimental design, execution and interpretation lower the enthusiasm for this manuscript.

Specific Comments

The mRNA of C9orf72 in humans exists in three transcript variant and these give rise to two protein isoforms (long and short C9orf72 protein). Does the zebrafish C9orf72 ortholog also exist in different transcript variants? If so, what are the mRNA levels of each of the transcripts? Similarly, the western blot shown to quantify C9orf72 protein only shows one band corresponding to C9orf72. Is this because only one isoform exists in zebrafish? Or because you are unable to detect the two protein isoforms?

There is a significant range in the survival of the C9-miR larvae and fish. Some larvae succumb to premature death 6-14 dpf, whilst others go onto become adult zebrafish that are then characterised. What causes this range in survival phenotypes? Is it correlated with the level of C9orf72 knock-down?

For all experiments, a better control than just WT zebrafish should be included to ensure that the phenotypes observed are not due to the transgenic DNA construct or egg injection. A better control would be to generate a second transgenic line using a scrambled/non-targeting miRNA construct.

The authors claim to observe cytoplasmic aggregation of TDP-43 in the zebrafish skeletal muscle. Firstly, the images provided are of poor quality, and it might be helpful to add a muscle-specific marker protein to provide an outline of the actual muscle, in addition to TDP-43 and Hoechst staining. Also, given that TDP-43 pathology is mostly seen in neurons in C9orf72 patients, the authors should also assess whether TDP-43 aggregation is seen in the motor neurons. Finally, the cytoplasmic inclusions need to be quantified, e.g. by using the commonly applied nucleo-cytoplasmic ratio measurements.

The NMJ structural integrity is assessed up to 6 dpf, yet many of the zebrafish have a longer lifespan. The reviewer is confused as to why this is assessed at such an early time point and would like to see the NMJ integrity assessed at several more timepoints, particularly in aged fish.

The authors claim that the reduction in SV2a seen in the C9-miRNA fish indicates that C9orf72 regulates synaptic vesicle exocytosis and synapse stability at the NMJ, however no mechanism has been described to back up this claim. Is this a direct effect? Do these proteins interact? How is the reduction in SV2a mediated by C9orf72?

The authors found reduced levels of Rab3a in the C9-knock down fish. This is in direct contrast to what has been seen in C9-knockout mice (Xiao et al., 2019, *Acta neuropath comm*s), where the levels of Rab3a were shown to remain stable. The authors should discuss this discrepancy in the discussion.

Line 256 – sentence incomplete / does not make sense currently

Figure 3c – The authors state that the large motor neurons are reduced in size in the C9-miRNA fish. This could be a very observational finding thus quantification of the motor neuron size is needed to support this claim.

Figure 4 – These are poor quality images. Hoechst is spelt incorrectly.

Figure 4b – The authors should include images of the TDP-43 and Hoechst staining in WT/control fish as a comparison. Furthermore, the mislocalisation of TDP-43 should be quantified to confirm there is a significant effect.

Reviewer #2 (Remarks to the Author):

Hexanucleotide repeat expansions in C9orf72 are the most common known genetic cause of amyotrophic lateral sclerosis (ALS), but the mechanism of action remains unclear. Both gain- as well as loss-of-function models have been proposed and even combinations of both. Although one zebrafish study showed that morpholino knockdown of zC9orf72 induced axon degeneration and motor dysfunction (Ciura et al., 2013), multiple groups have reported that knocking out C9orf72 from mice show immune-related effects but no detectable neurodegeneration (Stepto et al., 2014). Here, Butti et al. report a new miRNA-based knockdown of zC9orf72 induces ALS-like phenotypes, supporting a possible loss-of-function model. The range of phenotypes observed is impressive and their clear link to ALS is impressive. Thus, this is a very interesting report, but more work needs to be done before the manuscript is ready for publication.

One important point is to put the findings into the larger context. As indicated above, multiple teams have already reported knockouts of C9orf72 in mice that are quite different from the results presented here. How do the authors reconcile these differences, particularly given that mice are evolutionarily closer to human ALS patients compared to zebrafish?

A critical technical point is to demonstrate that the effects are due to zC9orf72 reduction and not off-target effects. For example, does transcriptome analysis show other genes reduced? Can the phenotypes be rescued by overexpression? This is particularly important because, from what I can see, the control seems to simply be wild type animals instead of scrambled miRNA. Since the findings seem to be quite different (and impressive) compared to the mouse knockout, it is critical to be clear that the phenotypes are due to the intended target.

Other, more minor points, include:

- Scale bar is missing in Fig 1B
- Please include the complete western blot for Fig 1D as supplementary data (not just cropped bands)
- Please describe statistical test used for p-values in all figure legends. The Methods has a paragraph listing multiple tests, but it isn't clear which test was used for each individual analysis.
- Why do control animals show reduced survival at day 15 in Fig 2C?
- Please include age of animals in legend for Fig 2D.
- It is difficult to follow the logic of dsRED in Fig 1A versus use of GFP in Fig 1B. Please make a better connection between the two.
- How old were fish used in Fig 3?
- Please show a graph for the ChAT reduction of 19.2% on page 7, line 166
- Please include a scale bar in Figure 3C
- Why are such young animals characterized in figure 4, 5, and 6? Wouldn't it be useful to include an analysis of adult animals to continue to analysis of Figure 3?
- Quantify nuclear vs cytoplasmic TDP-43 in Figure 4. Very hard to see if TDP-43 is nuclear in Fig. 4A.

- What about TDP-43 in MN soma in the spinal cord? Do spinal MN nuclei show nuclear depletion as in ALS?
- Please include scale bars in Figs 5A and C.
- Table S2 not included in manuscript
- No error bars are included on Fig 7. How many replicates performed for proteomics? There should be more than just a single replicate
- I don't really see the value of Figs 7B-E, where the prominent terms are quite generic, for example "cell part" and "catalytic activity". Maybe just delete this? Can it better integrated into the storyline?

Ciura, S., S. Lattante, I. Le Ber, M. Latouche, H. Tostivint, A. Brice, and E. Kabashi. 2013. Loss of function of C9orf72 causes motor deficits in a zebrafish model of amyotrophic lateral sclerosis. *Ann Neurol* 74:180-187.

Stepito, A., J.M. Gallo, C.E. Shaw, and F. Hirth. 2014. Modelling C9ORF72 hexanucleotide repeat expansion in amyotrophic lateral sclerosis and frontotemporal dementia. *Acta Neuropathol* 127:377-389.

Reviewer #3 (Remarks to the Author):

Review of manuscript by Patten et al. MS#COMMSBIO-20-2120

This manuscript describes the generation of a transgenic zebrafish model with reduced expression of C9orf72, and effects of this mutation on mortality and motor function. Mutated fish exhibit motor deficits, and abnormalities are reported in the histology/histochemistry of skeletal muscle and its innervation. Most mutated fish die within 15 days post-fertilization. C9orf72 abnormalities are linked to the development of amyotrophic lateral sclerosis (ALS) and fronto-temporal dementia (FTD) in humans, and the authors conclude that their study demonstrates a novel role for C9orf72 in the pathogenesis of ALS/FTD.

The study employs an impressive range of experimental techniques, but several flaws/omissions cast doubt on many of the reported findings. One problem is inadequate controls for off-target effects of the miRNA. Such controls might include scrambled RNA with the same bases, and rescue experiments (as described in Seok et al. 2018 *Cell Mol Life Sci* 75:797 PMID: 28905147). Experiments knocking out C9orf72 might also be more convincing in testing the effects of decreasing C9orf72 function.

Another problem is that it is unclear whether the reported motor findings were the primary cause of, or merely a secondary result of, other factors leading to the early death of the mutated fish. Some studies involved adult fish, a tiny minority because only 2-5% of the mutated fish survived beyond 15 days post-fertilization. No systems other than motor neurons/skeletal muscle were examined – what about cardiovascular or gastrointestinal systems, or the immune system, which is deficient in some mutant C9orf72 mice? Could the reported presynaptic changes have resulted simply from motor neuron degeneration?

The results are often described and illustrated in such a sketchy fashion that this reader was not convinced about many of the reported findings. For example, in Fig. 3c, how were the ChAT-positive motor neurons identified and measured in the midst of so much apparently non-specific staining? Concerning Fig. 4, why was side-by-side TDP-43 and Hoescht staining not illustrated for controls? In Fig. 5 the micrographs are shown at such low magnification that it is impossible to see how the SV2/ α -bungarotoxin puncta were analyzed, and no information is given concerning the number of puncta counted and precautions taken to ensure objective selection of fields and bias-free counting. Similar concerns apply to the Rab3A and FM-143 measurements in Fig. 8. The electrophysiological recordings of miniature end-plate currents (NOT calcium currents as stated on p. 18) are illustrated only at a very

slow time scale, with no information concerning objective criteria used to detect these currents and measurement of amplitude and time course. Also, how would the hypothesized defects in presynaptic transmission relate to the observed reduction in muscle fiber diameter?

The manuscript would benefit from careful editing – there are many undefined acronyms (GEF, DPR, DENN, HBSS, DEP, RTo), with some methods sections reading like a lab notebook. Also, summary statements need to be less expansive, and more closely tied to the data presented.

In sum, the authors report that the reduced expression of C9orf72 in their novel zebrafish model results in loss-of-function in presynaptic aspects of neuromuscular transmission. But the rapid, high mortality associated with the mutation raises doubts concerning non-specific effects of the miRNA, and/or whether the reported defects are primary or secondary to other effects of the mutation. And the presynaptic defects

themselves need to be more carefully documented.

Point-by point response to the reviewers:

We would like to begin by thanking the reviewers for their constructive comments. We have made substantial changes to the manuscript to address their concerns, with all significant changes tracked in the revised Ms Word document. A point-by-point response to the reviewers' comments is provided below.

Reviewer #1 (Remarks to the Author):

In this paper Butti et al. describe the generation and characterisation of transgenic zebrafish line whereby miRNA is used to knock-down the C9orf72 gene. The authors show that reduced C9orf72 function leads to motor defects, muscle atrophy, motor neuron loss and premature death. They then analyse the structure and functional capacity of the neuromuscular junctions (NMJs) and found that the C9-miRNA zebrafish have reduced functional NMJs from 6 dpf. Finally, the authors perform proteomic analysis and find a downregulation of synaptic proteins in the C9-miRNA fish. Investigation of synaptic vesicle cycling at the NMJ indicates C9-miRNA fish have slowed exocytotic activity of synaptic vesicles and conclude that C9orf72 plays a key role in regulating pre-synaptic vesicle release at the NMJ.

While this is an interesting finding, there is lack of novelty as several other prior studies have already concluded that C9orf72 protein plays a role in vesicle cycling. In addition, numerous concerns with the experimental design, execution and interpretation lower the enthusiasm for this manuscript.

- We wish to thank the reviewer for their comments and the valuable suggestions on how to improve our manuscript. We agree with the reviewer that there are several studies reporting a role of C9orf72 in vesicle trafficking. However, these studies are primarily focused on endocytotic and autophagic vesicles, implicating C9orf72 function in endosomal trafficking and autophagy and not at the synapses. Xiao et al., 2019 recently reported that C9orf72 is expressed presynaptically and postsynaptically¹. Frick and colleagues also reported that C9orf72 is localized to presynapses². However, the physiological function of C9orf72 at synapses, particularly presynaptically, remains largely unexplored. Here we reveal an important and novel role of C9orf72 in synaptic vesicle release at neuromuscular junctions.

-Importantly, another major novelty of our manuscript is that we have developed a C9orf72 loss of function model that replicates major hallmark of ALS, in particular TDP-43 pathology, indicating a mechanistic link between C9orf72 and TDP-43 in determining the cellular localization of TDP-43. We believe that the experiments carried out during the revision substantially strengthened the manuscript.

Specific Comments

The mRNA of C9orf72 in humans exists in three transcript variant and these give rise to two protein isoforms (long and short C9orf72 protein). Does the zebrafish C9orf72 ortholog also exist in different transcript variants? If so, what are the mRNA levels of each of the transcripts? Similarly, the western blot shown to quantify C9orf72 protein only shows one band corresponding to C9orf72. Is this because only one isoform exists in zebrafish? Or because you are unable to detect the two protein isoforms?

Ensembl BLAST/BLAT | VEP | Tools | BioMart | Downloads | Help & Data | Blog

Zebrafish (GRCz11) Location: 18:12,781,787-18:12,778,963 Gene: C13H9orf72 Transcript: C13H9orf72.2

Gene-based displays

- Summary
- Splice variants
- Transcript comparison
- Gene aliases
- Sequences
- Secondary structure
- Comparative Genomics
- Genomic alignments
- Gene tree
- Gene families tree
- Orthologues
- Paralogues
- Enzyme protein families
- Orthologues
- GO: Biological process
- GO: Molecular function
- GO: Cellular component
- Phenotypes

Gene: C13H9orf72 ENSDARG00000011837

Description: agc100M6 (Source:ZFIN; Acc:ZDB-GENE-240811-47-49)

Gene Synonyms: C9orf72, c13h9orf72

Location: Chromosome 13: 12,781,787-12,778,967 forward strand. GRCz11:CM000287.2

About this gene: This gene has 1 transcript (other variants are orthologous and is associated).

Transcripts

Show/hide columns (1 hidden)

Name	Transcript ID	bp	Protein	Biotype	UniProt Match	Flags
C13H9orf72.201	ENSDART00000015127.2	2427	485aa	Protein coding	OCLT:84.6	APR05 PT

Image response 1

-Zebrafish has only 1 C9orf72 orthologue (zgc:100846; c13h9orf72) and giving rise to one protein-coding transcript (ENSDART00000015127.7; Image response 1). A single band specific the size of the similar to the human C9orf72 long-transcript was observed in our Western Blot analysis. We now include the full blot in Supplementary Fig. 1a.

There is a significant range in the survival of the C9-miR larvae and fish. Some larvae succumb to premature death 6-14 dpf, whilst others go on to become adult zebrafish that are then characterised. What causes this range in survival phenotypes? Is it correlated with the level of C9orf72 knock-down?

- Thank-you for this comment. Indeed, due to the nature of the knockdown technology used, animal should present some degree of variability regarding the level of the transgene expression, hence a small variability in term of C9orf72 knockdown. A range of survival phenotypes at larval and adult stages are often observed in zebrafish models of ALS³⁻⁶. Zebrafish laboratory strains are not highly inbred owing to the negative impact of inbreeding on reproduction⁷. Thus, the time between disease onset and death, which is very stereotyped in some inbred animal models like mice lines, is much more variable in zebrafish C9-miR (and in other ALS models³⁻⁶) likely due to genetic heterogeneity within strains.

For all experiments, a better control than just WT zebrafish should be included to ensure that the phenotypes observed are not due to the transgenic DNA construct or egg injection. A better control would be to generate a second transgenic line using a scrambled/non-targeting miRNA construct.

- We agree with the reviewer. To ensure that the phenotypes observed in the zebrafish C9orf72-LOF model are specific to C9orf72 knockdown, we have performed rescue experiments. We now report in this revised manuscript that the key phenotypes: motor behavioural defects, NMJ anomalies and TDP-43 pathology in C9-miR fish can be significantly rescued by C9orf72 mRNA (Fig. 2 and 4; Line 158-161; Line 177-178 and Line 212-213). Altogether, these results strengthen the conclusions of the manuscript.

Noteworthy, we have considerable experience with zebrafish egg microinjection experiments. We are also studying additional ALS risk-genes using the same transgenic RNAi such as NEK1, TBK1 and GGNBP2, as well as non-ALS related genes such as NRXN1a and NRXN1b. None of the transgenic lines demonstrates similar phenotype as described here in our C9-miR line, demonstrating strong specificity of our findings.

The authors claim to observe cytoplasmic aggregation of TDP-43 in the zebrafish skeletal muscle. Firstly, the images provided are of poor quality, and it might be helpful to add a muscle-specific marker protein to provide an outline of the actual muscle, in addition to TDP-43 and Hoechst staining. Also, given that TDP-43 pathology is mostly seen in neurons in C9orf72 patients, the authors should also assess whether TDP-43 aggregation is seen in the motor neurons. Finally, the cytoplasmic inclusions need to be quantified, e.g. by using the commonly applied nucleo-cytoplasmic ratio measurements.

- Thank-you for this comment. We apologize for the poor quality of the images in the original submission. These images were replaced with high-quality images (Fig. 4b) along with quantification of the cytoplasmic inclusions (Fig. 4c) as suggested by the reviewer. We have also assessed TDP-pathology in spinal motor neurons (Fig. 6) and observed nuclear depletion of TDP-43 in C9-miR fish compared to controls (Fig. 6; Line 242-255)

- Although there are differences in the immunohistochemistry protocols for optimal staining for each antibody, we have attempted to adjust the protocols to perform co-staining with TDP-43 and Phalloidin (muscle marker). However, we had difficulty getting these to work. To take the reviewer's comment into considerations, we included Fig 4a to illustrate the morphology of the large nuclei in zebrafish skeletal muscles for non-zebrafish experts to help them understand the structures and stainings in Fig. 4b.

The NMJ structural integrity is assessed up to 6 dpf, yet many of the zebrafish have a longer lifespan. The reviewer is confused as to why this is assessed at such an early time point and would like to see the NMJ integrity assessed at several more timepoints, particularly in aged fish.

- This 6 dpf age was chosen to perform an exhaustive analysis of NMJ structural integrity because it corresponds to a stage when the motor behavioural phenotype is distinct. Additionally, one of the earliest hallmarks of disease in mouse models of ALS is NMJ defects that are seen as early as one to two months of age, before overt phenotypes⁸⁻¹⁰. To take the reviewer's comment into consideration, we have now included data on the assessment of NMJ integrity in aged fish (Fig. 5c,d; Line 225-230).

The authors claim that the reduction in SV2a seen in the C9-miRNA fish indicates that C9orf72 regulates synaptic vesicle exocytosis and synapse stability at the NMJ, however no mechanism has been described to back up this claim. Is this a direct effect? Do these proteins interact? How is the reduction in SV2a mediated by C9orf72?

- Thanks for this comment. The conclusion that C9orf72 regulates synaptic vesicle exocytosis and synapse stability at the NMJ is made from strong evidences that (1) synaptic currents (mEPC quantal release) at NMJ are impaired in C9-miR fish; (2) FM1-43 loading in presynaptic terminals is significantly reduced in C9-miR animals; (3) reduction of SV2 and Rab3+ puncta in C9-miR fish. These data correlate well with our proteomic analysis, demonstrating a reduction in SV2a protein, an important synaptic protein involved exocytotic release of neurotransmitters, in C9-miR animals. We agree with the reviewer that it is important to examine whether C9orf72 and SV2a interact. We have thus performed co-immunoprecipitation coupled with western blot assay to determine whether C9orf72 interacts with SV2a. Data from these extensive experimentations revealed that SV2a can be co-immunoprecipitated by C9orf72 (Fig. 7b; Line 284-288), evidencing that C9orf72 directly interacts with SV2a. Of important note, C9-rescue experiments significantly ameliorate the number of SV2 clustering at NMJs (Fig. 2c,d), suggesting that the interaction of C9orf72 with SV2a is likely important to stabilize SV2a protein expression. Altogether, these results provide a novel mechanistic link between SV2a and C9orf72 and they strengthen the mechanistic conclusions of the manuscript.

The authors found reduced levels of Rab3a in the C9-knock down fish. This is in direct contrast to what has been seen in C9-knockout mice (Xiao et al., 2019, Acta neuropath comm), where the levels of Rab3a were shown to remain stable. The authors should discuss this discrepancy in the discussion.

- To take the reviewer's point into consideration, we now include in the discussion section of this revised manuscript a comparison between the observations from Xiao et al., 2019^l and our findings and discuss the implications of this dataset (Line 382-388).

Line 256 – sentence incomplete / does not make sense currently

- Corrected, thanks

Figure 3c – The authors state that the large motor neurons are reduced in size in the C9-miRNA fish. This could be a very observational finding thus quantification of the motor neuron size is needed to support this claim.

- We thank the reviewer for bringing this to our attention and making us realize that we should provide a better description of the motor neurons in zebrafish spinal cord and why we focused on the large motor neurons (Line 231-234). We also provide the quantification of reduced motor neuron size in the C9-miR fish compared to controls (Fig. 5f, Line 234-236).

Figure 4 – These are poor quality images. Hoechst is spelt incorrectly.

- *This figure has now been replaced with new high-quality images (Fig. 4b).*

Figure 4b – The authors should include images of the TDP-43 and Hoechst staining in WT/control fish as a comparison. Furthermore, the mislocalisation of TDP-43 should be quantified to confirm there is a significant effect.

- *We agree with the reviewer. We now provide a new Fig. 4b that includes both WT control and C9-miR images of TDP-43 and Hoechst staining as well as the quantification of the mislocalisation of TDP-43 (Fig. 4c).*

Reviewer #2 (Remarks to the Author):

Hexanucleotide repeat expansions in C9orf72 are the most common known genetic cause of amyotrophic lateral sclerosis (ALS), but the mechanism of action remains unclear. Both gain- as well as loss-of-function models have been proposed and even combinations of both. Although one zebrafish study showed that morpholino knockdown of zC9orf72 induced axon degeneration and motor dysfunction (Ciura et al., 2013), multiple groups have reported that knocking out C9orf72 from mice show immune-related effects but no detectable neurodegeneration (Stepito et al., 2014). Here, Butti et al. report a new miRNA-based knockdown of zC9orf72 induces ALS-like phenotypes, supporting a possible loss-of-function model. The range of phenotypes observed is impressive and their clear link to ALS is impressive. Thus, this is a very interesting report, but more work needs to be done before the manuscript is ready for publication.

- We wish to thank the reviewer for the positive comments and appreciation of the significance of our work.

One important point is to put the findings into the larger context. As indicated above, multiple teams have already reported knockouts of C9orf72 in mice that are quite different from the results presented here. How do the authors reconcile these differences, particularly given that mice are evolutionarily closer to human ALS patients compared to zebrafish?

- We agree with the reviewer and we now include in the discussion section of this revised manuscript a thorough comparison between results from C9orf72 knockouts in mice and our C9orf72 knockdown model (Line 326-341).

A critical technical point is to demonstrate that the effects are due to zC9orf72 reduction and not off-target effects. For example, does transcriptome analysis show other genes reduced? Can the phenotypes be rescued by overexpression? This is particularly important because, from what I can see, the control seems to simply be wild type animals instead of scrambled miRNA. Since the findings seem to be quite different (and impressive) compared to the mouse knockout, it is critical to be clear that the phenotypes are due to the intended target.

- Thanks for this comment. We agree with the reviewer. To ensure that the phenotypes observed in the zebrafish C9orf72-LOF model are specific to C9orf72 knockdown, we have performed rescue experiments. We now report in this revised manuscript that the key phenotypes: motor behavioural defects, NMJ anomalies and TDP-43 pathology in C9-miR fish can be significantly rescued by C9orf72 mRNA (Fig. 2 and 4; Line 158-161; Line 177-178 and Line 212-213). Altogether, these results strengthen the conclusions of the phenotypes in C9-miR fish being specifically due to a reduction of the zebrafish C9orf72.

Other, more minor points, include:

- Scale bar is missing in Fig 1B

- We thank the reviewer for catching this missing scale bar, which we have now added in Fig. 1b.

- Please include the complete western blot for Fig 1D as supplementary data (not just cropped bands)

- We have included the complete western blot in Supplementary Fig. 1a.

- Please describe statistical test used for p-values in all figure legends. The Methods has a paragraph listing multiple tests, but it isn't clear which test was used for each individual analysis.

- Thanks for this comment. We have ensured that all the statistical test performed are stated in the figure legends.

- Why do control animals show reduced survival at day 15 in Fig 2C?

- It is typical to have a reduction (70-80%) in survival rate in control fish around 10 to 15 days in experimental procedures (petri dishes). There is a critical period between 8 to 9 days when zebrafish larvae have consumed the nutrients stored in their yolk so that catabolism is no longer sufficient to support growth and survival. The transition from yolk sac nutrients to larval feeding results in a decline survival rate in experimental procedures as well as in their natural environment/housing tanks^{11,12}.

- Please include age of animals in legend for Fig 2D.

- Thanks for this comment. We have ensured that age of animals is stated in all the figure legends.

- It is difficult to follow the logic of dsRED in Fig 1A versus use of GFP in Fig 1B. Please make a better connection between the two.

- We thank the reviewer for catching this error. We have updated Fig. 1a to include the GFP in the construct, which now matches the text description and provides a better connection to Fig. 1b.

- How old were fish used in Fig 3?

- As stated above, we have ensured that age of animals is stated in all the figure legends.

- Please show a graph for the ChAT reduction of 19.2% on page 7, line 166

- We now provide the quantification of reduced motor neuron (ChAT+) size in the C9-miR fish compared to controls (Fig. 5f, Line 234-236).

- Please include a scale bar in Figure 3C

- We thank the reviewer for catching this missing scale bar, which we have now added. Of note, Figure 3C is now Fig. 5e.

- Why are such young animals characterized in figure 4, 5, and 6? Wouldn't it be useful to include an analysis of adult animals to continue to analysis of Figure 3?

- To gain insight into the function of C9orf72, young animals at 6 dpf were examined. This age was chosen to perform an exhaustive analysis of NMJ structural integrity and function because it corresponds to a stage when the motor behavioural phenotype is distinct. Additionally, to understand the role of C9orf72, we have capitalized on the technical advantages of zebrafish and performed in vivo experiments in whole-animal such as electrophysiological recordings, whole fish imaging and FMI-43 loading. Such experiments are technically difficult or impossible to perform in aged fish. Of note, we have now included data on the assessment of NMJ integrity in aged fish (Fig. 5c,d; Line 225-230), which is feasible on fish sections.

- Quantify nuclear vs cytoplasmic TDP-43 in Figure 4. Very hard to see if TDP-43 is nuclear in Fig. 4A.

- Thank-you for this comment. These images were replaced with high-quality images (Fig. 4b) along with quantification of the cytoplasmic inclusions (Fig. 4c) as suggested by the reviewer.

- What about TDP-43 in MN soma in the spinal cord? Do spinal MN nuclei show nuclear depletion as in ALS?

- Thank-you for this comment. We have assessed TDP-pathology in spinal motor neurons (Fig. 6) and observed nuclear depletion of TDP-43 in C9-miR fish compared to controls (Fig. 6; Line 242-255).

- Please include scale bars in Figs 5A and C.

- We thank the reviewer for catching these missing scale bars, which we have now added

- Table S2 not included in manuscript

- Table S2 is now included in the revised manuscript.

- No error bars are included on Fig 7. How many replicates performed for proteomics? There should be more than just a single replicate

- Indeed, proteomic analyses were performed in four experimental replicates per genotype. We have updated the Figure to include the error bars. Thank you for this comment.

- I don't really see the value of Figs 7B-E, where the prominent terms are quite generic, for example "cell part" and "catalytic activity". Maybe just delete this? Can it better integrated into the storyline?

- We understand the reviewer's point. These terms are automatically generated by the PANTHER analysis software upon performing the functional protein enrichment clustering analysis of the dysregulated proteins in our dataset. We have moved these figures to the Supplementary Figures.

Reviewer #3 (Remarks to the Author):

Review of manuscript by Patten et al. MS#COMMSBIO-20-2120

This manuscript describes the generation of a transgenic zebrafish model with reduced expression of C9orf72, and effects of this mutation on mortality and motor function. Mutated fish exhibit motor deficits, and abnormalities are reported in the histology/histochemistry of skeletal muscle and its innervation. Most mutated fish die within 15 days post-fertilization. C9orf72 abnormalities are linked to the development of amyotrophic lateral sclerosis (ALS) and fronto-temporal dementia (FTD) in humans, and the authors conclude that their study demonstrates a novel role for C9orf72 in the pathogenesis of ALS/FTD.

The study employs an impressive range of experimental techniques, but several flaws/omissions cast doubt on many of the reported findings. One problem is inadequate controls for off-target effects of the miRNA. Such controls might include scrambled RNA with the same bases, and rescue experiments (as described in Seok et al. 2018 Cell Mol Life Sci 75:797 PMID: 28905147). Experiments knocking out C9orf72 might also be more convincing in testing the effects of decreasing C9orf72 function.

- We wish to thank the reviewer for the positive comments and appreciation of the significance of our work. We agree with the reviewer and to ensure that the phenotypes observed in the zebrafish C9orf72-LOF model are specific to C9orf72 knockdown, we have performed rescue experiments. We now report in this revised manuscript that the key phenotypes: motor behavioural defects, NMJ anomalies and TDP-43 pathology in C9-miR fish can be significantly rescued by C9orf72 mRNA (Fig. 2 and 4; Line 158-161; Line 177-178 and Line 212-213). Altogether, these results strengthen the conclusions of the manuscript. We believe that the experiments carried out during the revision substantially strengthened the manuscript.

Another problem is that it is unclear whether the reported motor findings were the primary cause of, or merely a secondary result of, other factors leading to the early death of the mutated fish. Some studies involved adult fish, a tiny minority because only 2-5% of the mutated fish survived beyond 15 days post-fertilization. No systems other than motor neurons/skeletal muscle were examined – what about cardiovascular or gastrointestinal systems, or the immune system, which is deficient in some mutant C9orf72 mice? Could the reported presynaptic changes have resulted simply from motor neuron degeneration?

- Thank-you for this comment. In this revised manuscript, we have performed rescue experiments and provide strong evidences that the motor deficits and NMJ defects in C9-miR fish are specific to C9orf72 knockdown (Fig. 2 and Fig. 4).

- During the characterization of our model, we focused on systems relevant to ALS (i.e. motor neurons and skeletal muscles) and dissected in details the role of loss of function of C9orf72. Of important note, the other systems that were deficient in C9orf72 knock-out mice, that is upon complete loss of C9orf72 deficiency unlike our model which present a partial loss of C9orf72 (i.e. knockdown). Additionally, complete deletion of C9orf72 does not occur in C9orf72 ALS/FTD patients. Strikingly, unlike the C9orf72 mice models, our mutant line replicates an important hallmark of ALS that is TDP-43 pathology (Fig. 4 and Fig. 6). We provide in the discussion section of the revised manuscript a thorough comparison between results from C9orf72 mice and our C9orf72 knockdown model (Line 326-341). Motor neuron degeneration/loss occur in adult fish. We show presynaptic changes as of larval stages consistent with findings that NMJ defects precede motor neuron loss in ALS patients and animal models^{8,10}.

-We do not note any dysfunction/anomalies at the level of the cardiovascular, gastrointestinal systems or the immune systems in C9-miR fish. At the morphological level, the fish do not present

Image response 2: Gastrointestinal (GI) and Cardiovascular system morphology. (a) No gross anomalies (e.g smaller size) observed in GI tract (red square). Heart morphology (b; *) and cardiovascular system (c; Fli-GFP) are normal in C9-miR fish. Note heart defect is typically presented as cardiac edema in larval zebrafish but this is not observed here in (b) in C9-miR.

cardiovascular or gastrointestinal anomalies (Image response 2)- such as cardiac edema, faulty circulation reduced size of gastrointestinal area which are typical anomalies when these systems are deficient¹³⁻¹⁵. Importantly, no dysregulation of proteins involved in the immune system functioning and inflammatory responses was revealed in our proteomic analysis, including those observed dysregulated in C9orf72 mice such as TREM and interleukins (IL-10, IL-6)¹⁶. A full list of dysregulated proteins observed in proteomic analysis is provided in Table S2. If any defects in these systems were apparent, we would have certainly examined them further and reported them in the study - but this was not the case.

The results are often described and illustrated in such a sketchy fashion that this reader was not convinced about many of the reported findings. For example, in Fig. 3c, how were the ChAT-positive motor neurons identified and measured in the midst of so much apparently non-specific staining?

- Thank-you for this comment. We apologize some methods/quantifications were incomplete or missing in our original manuscript. We have extensively revised the description of the methods, quantification and improved the description of analyses specific to the zebrafish system in this revised manuscript.

- For instance, the ChAT staining is very specific and labels motor neurons of differences size within the grey matter of the spinal cord. We realized the confusion raised with the Figure we presented in the original manuscript with only two motor neurons illustrated with a dotted circle- our apologies. We now provide a better description of the varying motor neuron sizes in zebrafish, why we focused our analysis on a particular set of motor neurons (Line 231-234) and how they were identified for analysis (Line 572-573). Of note, these ChAT-positive motor neurons are typical analyzed in zebrafish model of ALS^{3,5,6}.

Concerning Fig. 4, why was side-by-side TDP-43 and Hoescht staining not illustrated for controls?

- Thanks for the comment. This Figure has been replaced with a complete Figure (Fig. 4b) which now includes representative images of TDP-43 and Hoechst staining in control, C9-miR and C9-Rescue animals.

In Fig. 5 the micrographs are shown at such low magnification that it is impossible to see how the SV2/ α -bungarotoxin puncta were analyzed, and no information is given concerning the number of puncta counted and precautions taken to ensure objective selection of fields and bias-free counting. Similar concerns apply to the Rab3A and FM-143 measurements in Fig. 8.

- Our apologies, images of SV2/ α -bungarotoxin puncta are now provided at higher magnification (Fig. 2c). We have extensively revised the methodology and we also provide a detailed description in the methodology of how the counting was performed (NMJ: Line 507-510; Rab3a: Line 638-641 and FM1-43: Line 594-596). Noteworthy, we have considerable experience in performing such quantitative analysis at NMJs in zebrafish, including in ALS models^{17,18}.

The electrophysiological recordings of miniature end-plate currents (NOT calcium currents as stated on p. 18) are illustrated only at a very slow time scale, with no information concerning objective criteria used to detect these currents and measurement of amplitude and time course.

- Thanks for this comment. We apologize that the methods/analysis of miniature end-plate currents (mEPCs) were incomplete or missing. We also thank the reviewer for catching this mistake of naming the currents. We now provide complete detailed methodology regarding the electrophysiological recordings (Line 599-609) and the analysis of mEPCs in the methods section (Line 611-624).

Also, how would the hypothesized defects in presynaptic transmission relate to the observed reduction in muscle fiber diameter?

- It is well-established that the function of NMJ is important for maintaining muscle mass¹⁹. Impairment of NMJ function (as observed in our model) will likely results in muscle weakness and the behavioural anomalies. Muscle disuses impairs signalling pathways leading to muscle atrophy^{19,20}. Of note, this is conclusion is outside of the scope of this study.

The manuscript would benefit from careful editing – there are many undefined acronyms (GEF, DPR, DENN, HBSS, DEP, RTo), with some methods sections reading like a lab notebook.

- Thanks for this comment. We have defined the acronyms where needed and we have revised the methods.

Also, summary statements need to be less expansive, and more closely tied to the data presented.

- We went over the summary statements in the introduction and discussion, with all due respect, we believe they are concise, contain the necessary information to summarize the data well.

In sum, the authors report that the reduced expression of C9orf72 in their novel zebrafish model results in loss-of-function in presynaptic aspects of neuromuscular transmission. But the rapid, high mortality associated with the mutation raises doubts concerning non-specific effects of the miRNA, and/or whether the reported defects are primary or secondary to other effects of the mutation. And the presynaptic defects themselves need to be more carefully documented.

- These concerns have been addressed as described above. We believe that the experiments carried out during the revision substantially strengthened the manuscript and eliminates non-specificity concerns of the miRNA. Of important note, rapid and high mortality as well as larval and adult disease phenotypes are often observed in zebrafish models of ALS^{3-6,17,21,22}. The synaptic defects at NMJs are carefully documented and at several levels using a broad range of experimental techniques. We provide strong evidences that (1) reduction of SV2 at NMJs specific to C9orf72 knockdown; (2) synaptic currents (mEPC quantal release) at NMJ are impaired in C9-miR fish; (3) FM1-43 loading in presynaptic terminals is significantly reduced in C9-miR animals; (4) reduction of Rab3+ puncta in C9-miR fish. These data correlate well with our proteomic analysis, demonstrating a reduction in SV2a protein, an important synaptic protein involved exocytotic release of neurotransmitters, in C9-miR animals. Additionally, we show that C9orf72 interact with SV2a (Fig. 7b) strengthening the mechanistic conclusions of the paper.

Reference:

- 1 Xiao, S., McKeever, P. M., Lau, A. & Robertson, J. Synaptic localization of C9orf72 regulates post-synaptic glutamate receptor 1 levels. *Acta Neuropathol Commun* **7**, 161, doi:10.1186/s40478-019-0812-5 (2019).
- 2 Frick, P. *et al.* Novel antibodies reveal presynaptic localization of C9orf72 protein and reduced protein levels in C9orf72 mutation carriers. *Acta Neuropathol Commun* **6**, 72, doi:10.1186/s40478-018-0579-0 (2018).
- 3 Shaw, M. P. *et al.* Stable transgenic C9orf72 zebrafish model key aspects of the ALS/FTD phenotype and reveal novel pathological features. *Acta Neuropathol Commun* **6**, 125, doi:10.1186/s40478-018-0629-7 (2018).
- 4 Swaminathan, A. *et al.* Expression of C9orf72-related dipeptides impairs motor function in a vertebrate model. *Hum Mol Genet* **27**, 1754-1762, doi:10.1093/hmg/ddy083 (2018).
- 5 Ramesh, T. *et al.* A genetic model of amyotrophic lateral sclerosis in zebrafish displays phenotypic hallmarks of motoneuron disease. *Dis Model Mech* **3**, 652-662, doi:10.1242/dmm.005538 (2010).
- 6 Da Costa, M. M. *et al.* A new zebrafish model produced by TILLING of SOD1-related amyotrophic lateral sclerosis replicates key features of the disease and represents a tool for in vivo therapeutic screening. *Dis Model Mech* **7**, 73-81, doi:10.1242/dmm.012013 (2014).
- 7 Guryev, V. *et al.* Genetic variation in the zebrafish. *Genome Res* **16**, 491-497, doi:10.1101/gr.4791006 (2006).
- 8 Frey, D. *et al.* Early and selective loss of neuromuscular synapse subtypes with low sprouting competence in motoneuron diseases. *J Neurosci* **20**, 2534-2542 (2000).
- 9 Schaefer, A. M., Sanes, J. R. & Lichtman, J. W. A compensatory subpopulation of motor neurons in a mouse model of amyotrophic lateral sclerosis. *J Comp Neurol* **490**, 209-219, doi:10.1002/cne.20620 (2005).
- 10 Fischer, L. R. *et al.* Amyotrophic lateral sclerosis is a distal axonopathy: evidence in mice and man. *Exp Neurol* **185**, 232-240, doi:10.1016/j.expneurol.2003.10.004 (2004).
- 11 Hernandez, R. E., Galitan, L., Cameron, J., Goodwin, N. & Ramakrishnan, L. Delay of Initial Feeding of Zebrafish Larvae Until 8 Days Postfertilization Has No Impact on Survival or Growth Through the Juvenile Stage. *Zebrafish* **15**, 515-518, doi:10.1089/zeb.2018.1579 (2018).
- 12 Best, J., Adatto, I., Cockington, J., James, A. & Lawrence, C. A novel method for rearing first-feeding larval zebrafish: polyculture with Type L saltwater rotifers (*Brachionus plicatilis*). *Zebrafish* **7**, 289-295, doi:10.1089/zeb.2010.0667 (2010).
- 13 Miura, G. I. & Yelon, D. A guide to analysis of cardiac phenotypes in the zebrafish embryo. *Methods Cell Biol* **101**, 161-180, doi:10.1016/B978-0-12-387036-0.00007-4 (2011).
- 14 Wallace, K. N. & Pack, M. Unique and conserved aspects of gut development in zebrafish. *Dev Biol* **255**, 12-29, doi:10.1016/s0012-1606(02)00034-9 (2003).
- 15 Cox, A. G. *et al.* Selenoprotein H is an essential regulator of redox homeostasis that cooperates with p53 in development and tumorigenesis. *Proc Natl Acad Sci U S A* **113**, E5562-5571, doi:10.1073/pnas.1600204113 (2016).
- 16 O'Rourke, J. G. *et al.* C9orf72 is required for proper macrophage and microglial function in mice. *Science* **351**, 1324-1329, doi:10.1126/science.aaf1064 (2016).
- 17 Patten, S. A. *et al.* Neuroleptics as therapeutic compounds stabilizing neuromuscular transmission in amyotrophic lateral sclerosis. *JCI Insight* **2**, doi:10.1172/jci.insight.97152 (2017).
- 18 Vaccaro, A. *et al.* Methylene blue protects against TDP-43 and FUS neuronal toxicity in *C. elegans* and *D. rerio*. *PLoS One* **7**, e42117, doi:10.1371/journal.pone.0042117 (2012).
- 19 Tintignac, L. A., Brenner, H. R. & Ruegg, M. A. Mechanisms Regulating Neuromuscular Junction Development and Function and Causes of Muscle Wasting. *Physiol Rev* **95**, 809-852, doi:10.1152/physrev.00033.2014 (2015).
- 20 Lecker, S. H. *et al.* Multiple types of skeletal muscle atrophy involve a common program of changes in gene expression. *FASEB J* **18**, 39-51, doi:10.1096/fj.03-0610com (2004).

- 21 Armstrong, G. A. & Drapeau, P. Calcium channel agonists protect against neuromuscular dysfunction in a genetic model of TDP-43 mutation in ALS. *J Neurosci* **33**, 1741-1752, doi:10.1523/JNEUROSCI.4003-12.2013 (2013).
- 22 Armstrong, G. A. & Drapeau, P. Loss and gain of FUS function impair neuromuscular synaptic transmission in a genetic model of ALS. *Hum Mol Genet* **22**, 4282-4292, doi:10.1093/hmg/ddt278 (2013).

REVIEWERS' COMMENTS:

Reviewer #1 (Remarks to the Author):

In this resubmission, Butti et al. addressed many of the reviewers concerns and did thereby significantly improve on the quality of their manuscript. A couple of minor concerns remain that should be addressed.

Figure 4.

The reviewer is excited to see a quantification of the TDP-43 mislocalization. It is unfortunate that due to technical difficulties, the authors are unable to provide a co-labelling with a cellular marker to clearly visualize cytoplasmic TDP-43 aggregates – which makes the reviewer wonder how they measured the cytoplasmic TDP-43? Which guidelines were used to outline the cells to then obtain the cytoplasmic TDP-43 labeling intensity?

Finally – something strange is going on in the merged image of the C9-Rescue panel – the nuclear staining in the merged image does not look like the nuclear staining in the Hoechst panel?

Figure 6.

Again, the reviewer is glad to see the efforts to stain for TDP-43 in spinal cord sections. Unfortunately, the images provided are not supporting their conclusion, nor the quantification in panel C. TDP-43 is still present in the nucleus, but might be aggregated in one area of the nucleus? There is no clear depletion to be observed, and with N=1 image, it is hard to believe that there is indeed nuclear depletion in the SMNs. Perhaps the authors could provide additional images to support their quantification?

Reviewer #2 (Remarks to the Author):

The authors have addressed all points.

Reviewer #3 (Remarks to the Author):

The authors made most of the requested extensive revisions, and the manuscript is greatly improved. The specificity of the mutation was clarified with addition of rescue experiments, and the relationship between findings in zebrafish and mouse C9orf72 mutants is well explained. Immunohistochemical and fluorescence figures were revised and clarified, and the rationale for studying mutant zebrafish at different stages of the life cycle is better explained. This is a valuable multidisciplinary contribution to the research literature.

A few caveats:

-The electrophysiological records in Fig. 3 were unchanged; rise time and decay time cannot be appreciated from the records shown.

-The rationale for interpreting the Western blots in Fig. 7b could be explained more clearly, especially the IB, IP inputs on the left side.

A minor point in the Discussion: The sentence in lines 368-369 ("Rab3a plays an...") of the Discussion should be omitted because it duplicates the first 2 sentences at the beginning of that paragraph (lines 355-357).

line 233: meaning of "negative for both DAPI..."?

Line 235: omit "are"

Point-by point response to the reviewers:

We would like to begin by thanking the reviewers for their comments. We have addressed their minor concerns, with all significant changes tracked in the revised Ms Word document. A point-by-point response to the reviewers' comments is provided below.

Reviewer #1 (Remarks to the Author):

In this resubmission, Butti et al. addressed many of the reviewers concerns and did thereby significantly improve on the quality of their manuscript. A couple of minor concerns remain that should be addressed.

Figure 4. The reviewer is excited to see a quantification of the TDP-43 mislocalization. It is unfortunate that due to technical difficulties, the authors are unable to provide a co-labelling with a cellular marker to clearly visualize cytoplasmic TDP-43 aggregates – which makes the reviewer wonder how they measured the cytoplasmic TDP-43? Which guidelines were used to outline the cells to then obtain the cytoplasmic TDP-43 labeling intensity?

- A detailed description of the methodology used to quantify the nucleo-cytoplasmic distribution of TDP-43 in zebrafish skeletal muscles is provided in the methods section of the revised manuscript. These immunostaining experiments are performed in vivo in whole-mount zebrafish. Images were acquired using the same settings for all preparations and each image consisted only of zebrafish skeletal muscle structures. Of important note, zebrafish skeletal muscles are composed of large sarcoplasm and multiple nuclei (Fig. 4a). Our analyses were adapted from nucleo-cytoplasmic analyses of muscle preparations^{1,2}. In each image, using imageJ, we determined the background-corrected mean pixel fluorescence in a user-defined area. A threshold was used to define the nucleus and cytoplasm was defined as outside the nuclei. Nucleus was then detected using the ImageJ Plugin versatile ward tool and the remaining area within the user-defined region, devoid of nuclei, was considered as the large cytoplasm area and background-corrected mean pixel fluorescence in each area was measured. Strikingly, clusters of TDP-43, located outside of the nucleus of muscles cells, was only observed in C9-miR zebrafish, in addition to nucleo-cytoplasmic quantification, we also provided a quantification of the number of cytoplasmic TDP-43 clusters per muscle somite (i.e a defined area of the muscles) in control and C9-miR fish (Supplementary Figure 1b).

Finally – something strange is going on in the merged image of the C9-Rescue panel – the nuclear staining in the merged image does not look like the nuclear staining in the Hoechst panel?

- We wish to thank the reviewer for catching this error. We have replaced this Hoechst panel by the correct Hoechst panel that corresponds to the double staining images presented in this figure and which forms part of the original merged image.

Figure 6.

Again, the reviewer is glad to see the efforts to stain for TDP-43 in spinal cord sections. Unfortunately, the images provided are not supporting their conclusion, nor the quantification in panel C. TDP-43 is still present in the nucleus, but might be aggregated in one area of the nucleus? There is no clear depletion to be observed, and with N=1 image, it is hard to believe that there is indeed nuclear depletion in the SMNs. Perhaps the authors could provide additional images to support their quantification.

- We thank the reviewer for this comment. We do not believe that TDP-43 is aggregated in one area of the nucleus. We now provide a zoom image (Inset) in Fig. 6b to show that TDP-43 staining is also found outside of the nucleus (depicted by arrows). It appears that TDP-43 mislocalization is rather a dynamic process in our in vivo model and this likely accounts for the variability observed, with nuclear depletion more prominent in some fish than others. As suggested by the reviewer we added additional images to illustrate a more prominent case of nuclear depletion in the SMNs, supporting the quantification. Nevertheless, it is very clear from our analyses, that TDP-43 cellular distribution is perturbed in C9-miR compared to control fish.

Reviewer #3 (Remarks to the Author):

The authors made most of the requested extensive revisions, and the manuscript is greatly improved. The specificity of the mutation was clarified with addition of rescue experiments, and the relationship between findings in zebrafish and mouse C9orf72 mutants is well explained. Immunohistochemical and fluorescence figures were revised and clarified, and the rationale for studying mutant zebrafish at different stages of the life cycle is better explained. This is a valuable multidisciplinary contribution to the research literature.

- We wish to thank the reviewer for his/her comments and appreciation of the significance of our work.

A few caveats:

-The electrophysiological records in Fig. 3 were unchanged; rise time and decay time cannot be appreciated from the records shown.

- We understand the reviewer's point and we now provide representative images of an average mEPC event in Fig. 3b for control and C9-miR fish, where the rise time and decay time can be better appreciated.

-The rationale for interpreting the Western blots in Fig. 7b could be explained more clearly, especially the IB, IP inputs on the left side.

- In these experiments, in addition to our target protein (SV2a), we included both positive (SMRC8 a known interactor of C9orf72; IB and IP inputs on the left side) and negative controls. We now explain these better in the text result section (Line 288-290).

A minor point in the Discussion: The sentence in lines 368-369 ("Rab3a plays an...") of the Discussion should be omitted because it duplicates the first 2 sentences at the beginning of that paragraph (lines 355-357).

- This sentence (Rab3a plays an...) was removed to avoid duplication, thanks for this comment.

line 233: meaning of "negative for both DAPI"?

- It refers to no labelling upon DAPI staining, we have corrected this sentence to be more concise.

Line 235: omit "are"

- Thanks for catching this error, the word "are" has been removed.

Reference:

- 1 Liu, Y., Russell, S. J. & Schneider, M. F. Foxo1 nucleo-cytoplasmic distribution and unidirectional nuclear influx are the same in nuclei in a single skeletal muscle fiber but vary between fibers. *Am J Physiol Cell Physiol* **314**, C334-C348, doi:10.1152/ajpcell.00168.2017 (2018).
- 2 Brooks, N. E., Schuenke, M. D. & Hikida, R. S. Ageing influences myonuclear domain size differently in fast and slow skeletal muscle of rats. *Acta Physiol (Oxf)* **197**, 55-63, doi:10.1111/j.1748-1716.2009.01983.x (2009).